# Allosteric inhibition of PPM1D serine/threonine phosphatase via an altered conformational state

Peter G. Miller[1,2,10], Murugappan Sathappa[3,10], Jamie A. Moroco [3,10], Wei Jiang[3], Yue Qian[3], Sumaiya Iqbal [3], Qi Guo[3], Andrew O. Giacomelli [2,4], Subrata Shaw[3], Camille Vernier[3], Besnik Bajrami[3], Xiaoping Yang [2], Cerise Raffier[3], Adam S. Sperling [2,4,5], Christopher J. Gibson [2,4], Josephine Kahn[6], Cyrus Jin[7,8], Matthew Ranaghan [3], Alisha Caliman[3], Merissa Brousseau[3], Eric S. Fischer [7,8], Robert Lintner[2], Federica Piccioni[2], Arthur J. Campbell[2], David E. Root [2], Colin W. Garvie[3✉] & Benjamin L. Ebert [2,4,9✉]

*PPM1D* encodes a serine/threonine phosphatase that regulates numerous pathways including the DNA damage response and p53. Activating mutations and amplification of *PPM1D* are found across numerous cancer types. GSK2830371 is a potent and selective allosteric inhibitor of PPM1D, but its mechanism of binding and inhibition of catalytic activity are unknown. Here we use computational, biochemical and functional genetic studies to elucidate the molecular basis of GSK2830371 activity. These data confirm that GSK2830371 binds an allosteric site of PPM1D with high affinity. By further incorporating data from hydrogen deuterium exchange mass spectrometry and sedimentation velocity analytical ultracentrifugation, we demonstrate that PPM1D exists in an equilibrium between two conformations that are defined by the movement of the flap domain, which is required for substrate recognition. A hinge region was identified that is critical for switching between the two conformations and was directly implicated in the high-affinity binding of GSK2830371 to PPM1D. We propose that the two conformations represent active and inactive forms of the protein reflected by the position of the flap, and that binding of GSK2830371 shifts the equilibrium to the inactive form. Finally, we found that C-terminal truncating mutations proximal to residue 400 result in destabilization of the protein via loss of a stabilizing N- and C-terminal interaction, consistent with the observation from human genetic data that nearly all *PPM1D* mutations in cancer are truncating and occur distal to residue 400. Taken together, our findings elucidate the mechanism by which binding of a small molecule to an allosteric site of PPM1D inhibits its activity and provides insights into the biology of PPM1D.

[1] Center for Cancer Research, Massachusetts General Hospital, Harvard Medical School, Boston, MA, USA. [2] Broad Institute of MIT and Harvard University, Cambridge, MA, USA. [3] Center for the Development of Therapeutics, Broad Institute of MIT and Harvard University, Cambridge, MA, USA. [4] Department of Medical Oncology, Dana-Farber Cancer Institute, Harvard Medical School, Boston, MA, USA. [5] Division of Hematology, Department of Medicine, Brigham and Women's Hospital, Harvard Medical School, Boston, MA, USA. [6] Department of Internal Medicine, Brigham and Women's Hospital, Harvard Medical School, Boston, MA, USA. [7] Department of Cancer Biology, Dana-Farber Cancer Institute, Boston, MA, USA. [8] Department of Biological Chemistry and Molecular Pharmacology, Harvard Medical School, Boston, MA, USA. [9] Howard Hughes Medical Institute, Bethesda, MD, USA. [10]These authors contributed equally: Peter G. Miller, Murugappan Sathappa, Jamie A. Moroco. ✉email: cgarvie@broadinstitute.org; benjamin_ebert@dfci.harvard.edu

*P*PM1D encodes a serine/threonine phosphatase that negatively regulates multiple components of the DNA damage response (DDR)[1]. PPM1D activity is enhanced in numerous oncologic contexts via genetic amplification or C-terminal truncations causing loss of a degradation domain, both resulting in increased enzymatic function. Consistent with these findings, *PPM1D* overexpression in experimental models augments transformation through p53-dependent and independent mechanisms[2–4]. Small molecule inhibitors of PPM1D have been described, including a PPM1D-selective inhibitor, GSK2830371[5–11]. The selectivity for PPM1D was of particular interest, as the similarity between catalytic domains across PPM1D paralogs has posed challenges to developing inhibitors. GSK2830371 was found to bind to an allosteric site outside the catalytic domain, potentially explaining its selectivity, but the exact binding site and molecular mechanism of action remain unknown. Despite extensive efforts, the crystal structure of PPM1D has proven elusive[5].

Protein phosphatases are attractive therapeutic targets across numerous oncologic and non-oncologic indications. Of the nearly 200 known or predicted human phosphatases, many are mutated in human disease[12]. Phosphatases are broadly divided into tyrosine and serine/threonine phosphatases, each of which pose unique challenges for developing small molecule inhibitors. Phosphorylated tyrosine residues are highly charged, requiring effective competitive inhibitors to also be charged, resulting in unfavorable pharmacologic properties[13]. Serine/threonine kinases often work as holoenzymes and have highly conserved catalytic domains, making selective, competitive inhibition difficult[14]. As a result, the development of drugs targeting phosphatases has focused on noncompetitive, allosteric inhibitors for both tyrosine and serine/threonine phosphatases, with six small molecules currently in human trials[15,16].

We set out to define the mechanism by which GSK2830371 interacts with PPM1D and, in doing so, inform broader approaches to allosteric inhibition of serine/threonine phosphatases. As there is no crystal structure for PPM1D, we utilized computational modeling and functional mutagenesis approaches to define domain structures and residues required for PPM1D activity. We integrated these data with those from biochemical and biophysical assays, including hydrogen–deuterium exchange mass spectrometry (HDX-MS), to define the mechanism by which GSK2830371 binds, restructures, and inhibits PPM1D. These data also provide biological insights into human disease, including the distribution of *PPM1D* mutations found in cancer. Taken together, our data inform drug development efforts and provide a framework for a deeper understanding of allosteric inhibition of protein phosphatases.

## Results

**PPM1D homology model reveals distinct protein regions**. We created a homology model of PPM1D based on the published structures of human PPM1A, human PPM1B, and a PPM-type phosphatase from *A. Gambiae* (see "Methods")[17–19]. Due to a lack of structural information in the C-terminus, this model is truncated at residue 377, proximal to where most C-terminal truncating mutations occur. The modeled structure is comprised of two antiparallel β-sheets that form a β-sandwich at the core of the protein with α-helices packing against the outer faces. The catalytic site is located in the loops at one end of the β-sheets and contains two metal-binding sites coordinated directly by four acidic residues (D105, D192, D314, and D366) (Fig. 1A). Modeling of the structure revealed two regions that are unique to PPM1D, residues 39–95 and 219–295, which we will refer to as the loop and the flap region, respectively. Neither region is predicted to form well-defined secondary or tertiary structure. The

functional role of the loop is unknown but the flap region is proposed to be involved in substrate recognition[5,11,20]. The term flap has been used to describe regions at the catalytic site of phosphatases that are involved in catalysis and substrate recognition and have been observed to adopt more than one conformation[21,22]. Further comparison of PPM1D to its human paralogs revealed significant regional variation, with paralog conservation lowest in the loop, complete conservation at the metal-binding residues, and mixed conservation across the remainder of the protein (Fig. 1B, Supplementary Fig 1A, and Supplementary Tables 1A, B). We also observed a very low paralog conservation between residues 155–166, which we will refer to as the hinge, indicating that this region is unique to PPM1D.

Despite extensive efforts to crystallize the protein (Supplementary Table 2), we were unable to do so. Instead, we applied computational and functional genetic data to test the predictions of the homology model. First, we calculated the degree of evolutionary conservation across 19 mammalian orthologues and the degree of genetic constraint in the germline of the human population at each residue using the missense tolerance ratio (Fig. 1C, Supplementary Table 3)[23]. The model's predicted lack of structure in the loop is consistent with low evolutionary conservation and genetic constraint, whereas the predicted β-sheets in the N-terminus that precede the loop are conserved and constrained. While both the loop and the hinge are unique to PPM1D, the hinge is conserved and constrained, suggesting functional importance. The unstructured flap region shows variable paralog and ortholog conservation, as would be expected from a region that is predicted to mediate substrate recognition[20–22], and high genetic constraint underscoring its functional importance. Finally, the metal-binding residues, which interact with the magnesium cations, were integral for generating the model and are completely conserved and genetically constrained. Taken together, the evolutionary conservation data supports the predictions of the homology model.

**Saturation mutagenesis of PPM1D defines functional importance of each residue**. To interrogate the predictions from the homology model functionally at each residue, we performed a saturation mutagenesis screen of PPM1D, systematically mutating each amino acid to all alternative amino acids. First, we developed a cellular system to quantify PPM1D activity. We engineered the K562 cell line to express wild-type *TP53* from the endogenous locus, resulting in robust P53 activation and transcription following induction of DNA damage[24]. Next, we introduced *GFP* in frame into the endogenous *CDKN1A* (p21) locus at the C-terminus, resulting in the generation of a fusion p21-GFP gene, which allowed for quantification and selection of GFP-positive cells by flow cytometry after DNA damage induced by daunorubicin (Fig. 2A)[24]. As expected, overexpression of truncated *PPM1D* suppressed the induction of GFP (p21) whereas the introduction of catalytically inactivating mutations into *PPM1D* did not (Fig. 2B). Using this reporter system, we performed a saturation mutagenesis screen using a lentiviral library of 16,377 cDNAs encoding PPM1D$_{1–427}$ in which each amino acid is mutated to every other amino acid while the rest of the protein retains a wild-type sequence. The library includes stop codons and synonymous mutations as controls and utilizes 2- and 3-nucleotide changes for each codon change to minimize the likelihood that single nucleotide mutations introduced by PCR errors would be classified as true variants. After lentiviral transduction of the library into the reporter cells and selection in puromycin, daunorubicin was added and 24 h later the cells were sorted into GFP-hi and GFP-low fractions from which the representation of each PPM1D variant was quantified by

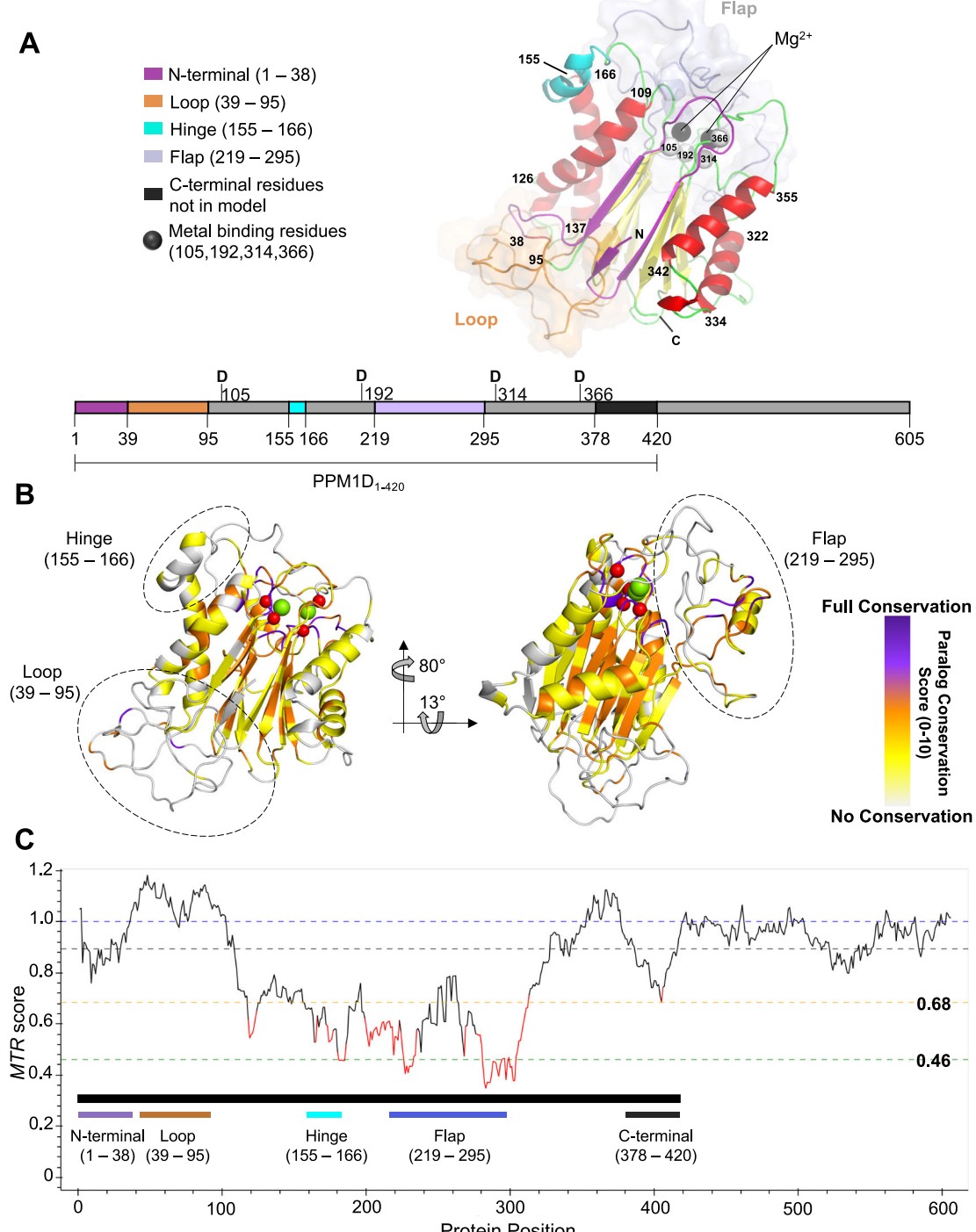

**Fig. 1 Generation of a homology model and use of computational genetics to characterize PPM1D. A** PPM1D homology model of residues 1–377 highlighting key structural regions of interest. The N-terminus (purple), loop (brown), hinge (cyan), and flap (light purple) are highlighted along with predicted alpha-helices (red) and beta-sheets (yellow), with the remainder of the protein in green. The loop and flap region are shown with an overlaying surface representation to emphasize the lack of structure predicted by the model. The metal-binding residues and metal cations are highlighted as spheres in black and white, respectively. A 2D representation is shown below the homology model. **B** PPM1D paralog values superimposed on the homology model. For each residue, the degree of identity across the human PPM1D paralogs was assessed and scored from 0 to 10 (purple represents full conservation, white represent no conservation). **C** Representation of genetic constraint in the human germline at each residue as calculate by the missense tolerance ratio. A ratio of less than 0.68 and 0.46 are in the bottom 25% and 5% of scores across the protein, respectively.

next-generation sequencing. For each variant we calculated the average of the log-fold change of representation in the GFP-low (retained PPM1D activity) and GFP-high (impaired PPM1D activity) populations (Fig. 2C, Supplementary Fig. 1B, C, Source data file: Saturation Mutagenesis Results)[25].

As expected, the synonymous mutations did not impact activity whereas stop codons showed clear loss of activity along the length of PPM1D, up to residue 400. From residue 400 onward the opposite was true, indicating that further C-terminal truncations can increase PPM1D activity, consistent with the

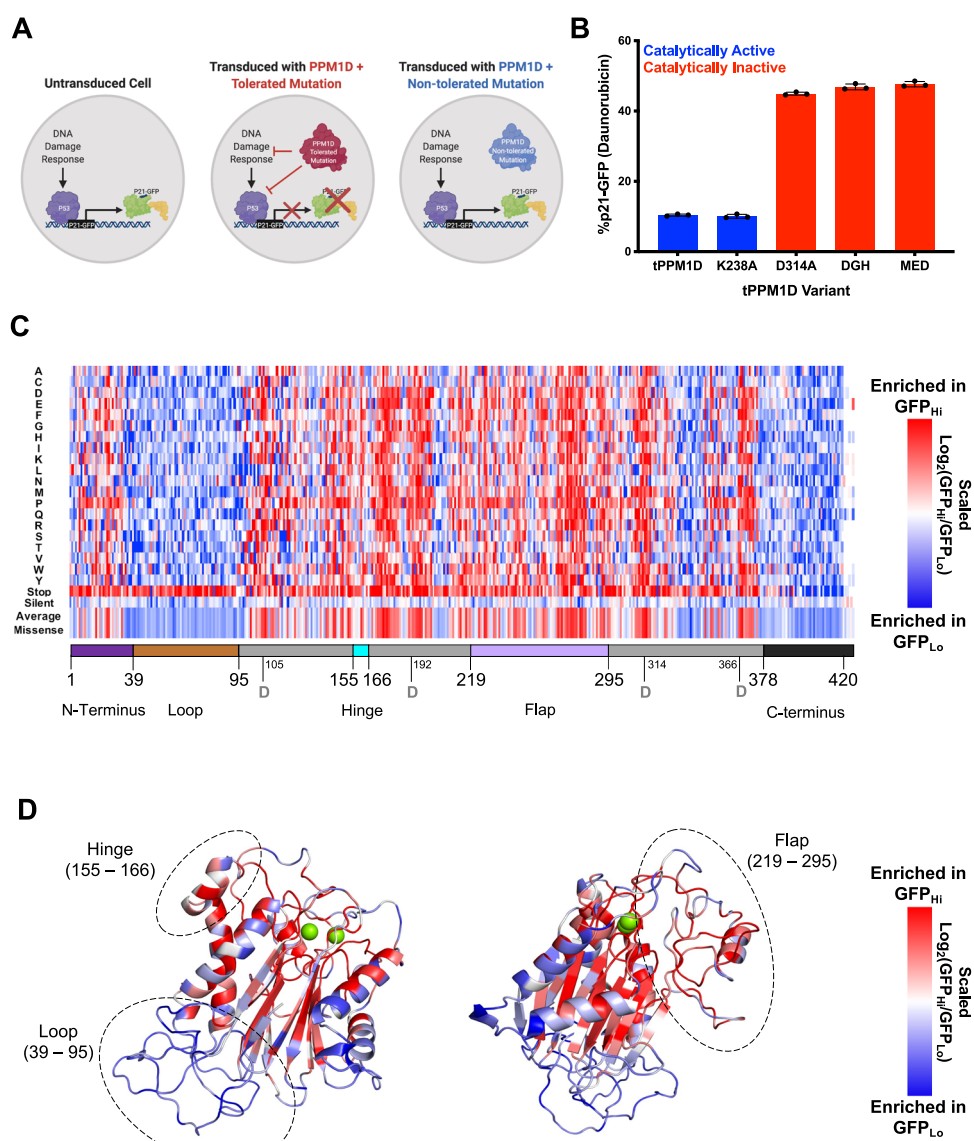

**Fig. 2 Deep mutational scanning identifies residues important for PPM1D function. A** Schematic of engineered K562 cells with wild-type *TP53* and GFP knocked into the C-terminus of the *CDKN1A* (p21) locus. In the presence of a DNA damaging agent, the DNA damage response (DDR) pathway induces p53 activation which transactivates the p21 locus, leading to expression of p21-GFP. After treatment with a DNA damaging agent such as daunorubicin, cells expressing a truncated PPM1D with a mutation that does not impair the function of the protein (red) suppresses the DDR and GFP expression whereas cells expressing truncated PPM1D that does impair protein function (blue) does not suppress the DDR or GFP expression. **B** Effect of mutations on PPM1D-mediated suppression of the DDR. Cells expressing truncated PPM1D (tPPM1D) or a mutation that does not affect protein function (tPPM1D-K238A) have lower p21-GFP levels at baseline and in the presence of daunorubicin. In contrast, cells expressing truncated PPM1D with mutations that impair catalytic function (tPPM1D-D314A, tPPM1D-DGH(105-107)→AAA, or tPPM1D-MED(21–23)→AAA)[17] have higher levels of p21-GFP at baseline and in the presence of daunorubicin. $N = 3$ biologically independent samples. Data are presented as mean values ± SEM. Source data are provided as a Source Data file. **C** Representation of deep mutagenesis screen. Each column represents the amino acid position and each row represents that amino acid change in the variant including stop codons, silent mutations, or the average of all the missense mutations. For each variant the $\log_2(GFP_{Hi}/GFP_{Lo})$ was calculated and scaled with red representing a greater ratio (impaired activity), and blue representing a lesser ratio (retained activity). See Supplemental Table 4 for source data. **D** Overlap of heatmap missense mutations on theoretical model. The average effect of the missense variants at each amino acid position are overlaid onto the theoretical model of truncated PPM1D.

somatic mutations observed in humans[3]. The N-terminal residues 1–31 are relatively intolerant to mutations, suggesting it is involved in structure and function. In contrast, the subsequent residues 31–96 are highly permissive to missense suggesting this region does not contribute significantly to structure/function and is consistent with the predicted lack of structure in the homology model. The remaining regions from residue 96 to approximately 376 show varying degrees of tolerance to missense mutations, though certain regions stand out as being particularly intolerant

which, apart from the hinge and 266–282, correspond closely with internal β-sheets predicted in the homology model. Region 266–282 is particularly interesting as it is located within the flap region of PPM1D that is proposed to be involved in substrate recognition and GSK2830371 binding[5]. Furthermore, residues predicted to be inward facing were more intolerant to mutations than those predicted to be solvent exposed, even within individual alpha-helices (Fig. 2D). Finally, the metal-binding residues and mutations known to impair catalytic activity were also intolerant

of mutations[17]. Thus, the functional saturation mutagenesis and computational genetic analyses highlight the functional relevance of the specific regions and secondary structures predicted by the PPM1D homology model.

**GSK2830371 functions as a non-competitive inhibitor and binds tightly at an allosteric site.** Having defined the domain structure of PPM1D, we next sought to determine the manner in which GSK2830371 inhibits PPM1D function. To examine the biochemical activity of GSK2830371, we used two independent enzymatic assays. The first assay, which uses the artificial substrate fluorescein diphosphate (FDP), provides a rapid, fluorescent-based readout of the activity of PPM1D and does not require substrate recognition (Supplementary Fig. 2A)[26]. The second, more physiologically-relevant assay uses liquid chromatography coupled to mass spectrometry (LC/MS) to detect dephosphorylation of T180 in full length p38 MAPK, a native substrate of PPM1D (Supplementary Fig. 2B, C). In both assays, GSK2830371 inhibited the enzymatic activity of $PPM1D_{1-420}$ with an $IC_{50}$ of approximately 6 nM and functioned as a non-competitive inhibitor predicted to bind to an allosteric site (Fig. 3A, B). GSK2830371 induced a thermal shift of approximately +8 °C in PPM1D by both differential scanning fluorimetry (DSF) and differential scanning calorimetry (DSC), indicating a strong stabilization of the protein due to compound binding (Fig. 3C and Supplementary Fig. 2D). This was confirmed by surface plasmon resonance (SPR) experiments which revealed a moderate on-rate ($3E^5$/s/M) and a slow-off-rate ($3E^{-4}$/s). The use of single-cycle kinetics was required to measure the true $K_d$ (1.8 nM) of the interaction, which was in agreement with the FDP and p38 MAPK $IC_{50}$ values (Fig. 3D).

As the relatively low solubility of GSK2830371 precluded isothermal titration calorimetry (ITC), we generated a structural analog of GSK2830371 (analog 1d) with comparable potency and binding affinity ($K_d = 2.7$ nM in SPR and IC50 = 22.4 nM in p38 MAPK assay) to GSK2830371 but significantly higher solubility (>100 μM) (Fig. 3E and Supplementary Fig. 3A–D). The ITC-defined $K_d$ of analog 1d on PPM1D was 10 nM and the interaction, favored both entropically and enthalpically, was primarily driven by entropic forces with a highly positive $\Delta S$ (25.8 cal/mol/K) (Fig. 3F), consistent with the presence of a hydrophobic interaction that drives an overall conformational change in the protein and/or water molecules. In aggregate, these studies indicate that GSK2830371 binds an allosteric site on PPM1D with high affinity and potentially induces a change in protein conformation.

**PPM1D exists in two conformations exhibited by movement of the flap region.** To investigate potential conformational changes in PPM1D caused by GSK2830371, as suggested by the ITC experiments, we utilized hydrogen deuterium exchange mass spectrometry (HDX-MS). This biophysical technique enables the evaluation of protein dynamics in solution by monitoring the incorporation of deuterium through exchange with amide hydrogens in the protein backbone[27,28]. Typically, unstructured and/or very mobile regions of the protein are highly deuterated at early labeling time points, while regions that are structured and/or buried within the core of the protein will take longer to incorporate deuterium.

We diluted recombinant $PPM1D_{1-420}$ into a deuterated buffer and measured the levels of deuterium uptake at time points ranging from 10 s to 4 h (see Supplementary Table 4 for a summary of HDX-MS experimental parameters). In agreement with the homology model, regions predicted to be less structured, to be solvent exposed, or to have increased conformational

flexibility, such as the loop and the C-terminal region, displayed enhanced deuterium uptake at early time points (Fig. 4A, B). Other regions associated with secondary structure showed less uptake at early time points, with a gradual increase in uptake over time. This included the N-terminal peptides spanning residues 5–12, supporting the homology model and experimental observations that N-terminal deletions of PPM1D lead to insoluble protein. Despite the predicted lack of structure in the flap region, there appeared to be a gradual increase in deuterium uptake in peptides spanning the flap (residues 219–295). Further inspection of the spectra for these some of these peptides revealed either peak-widening or a bimodal distribution, a phenomenon known as EX1 kinetics, which is lost in the typical centroid analysis of HDX-MS spectra[29,30]. Peptides exhibiting these characteristics are noted in Supplementary Data 1 (HDX-MS Deuterium Uptake Data) and Supplementary Data 2 (HDX-MS Deuterium Plots). The EX1 profile of a representative peptide (270–278, Fig. 4C) in this region suggests that the flap is sampling two different conformations; one solvent exposed and unrestrained by the rest of the protein, and another conformation that is more protected, possibly through interaction with the core of the protein.

To further investigate the effect of the flap and understand its interaction with the rest of the protein, we repeated the HDX-MS experiment to compare a flap-deletion construct, $PPM1D_{\Delta flap}$, in which the linker sequence GGSNGS replaced residues 219–287, to wild-type, truncated PPM1D ($PPM1D_{1-420}$). The $T_m$ of $PPM1D_{\Delta flap}$ was marginally higher than $PPM1D_{1-420}$ suggesting the deletion did not impact the tertiary structure significantly, and the removal of this dynamic region may have aided in protein stability (Supplementary Fig. 4A). Surprisingly, deletion of the flap region resulted in significantly more deuterium incorporation across several regions of the protein compared to wild-type, particularly in peptides spanning residues 101–115, 141–167, and 328–362 (Fig. 4D). This finding suggests that the flap region may interact with these regions, explaining the increase in deuterium uptake in its absence. The region between 152 and 167 encompasses the hinge, which we previously noted was unique to PPM1D and likely important for its structure or function. The differences observed at the boundaries of the flap deletion are likely due to reorganization of the local structural environment in the absence of the nearly 70 residues that make up the flap. Upon flap deletion, bimodal distributions were not observed elsewhere in the protein.

To determine whether there is a relationship between the flap region and the hinge, we replaced residues 155–166 with the sequence GGS ($PPM1D_{\Delta hinge}$) and compared the deuterium uptake with $PPM1D_{1-420}$. Analysis of the stability of $PPM1D_{\Delta hinge}$ showed a $T_m$ very similar to $PPM1D_{1-420}$ suggesting the tertiary structure was not significantly impacted by the deletion (Supplementary Fig. 4A). Remarkably, deletion of the hinge showed the exact opposite deuterium uptake profile of the flap deletion, with significantly less uptake in the same peptides spanning residues 101–115, 152–167, 328–362, and in the flap region itself (Fig. 4E). The apparent decrease in uptake in the flap region upon hinge deletion was due to a change in the bimodal distributions (Fig. 4C, Supplementary Data 1 and 2). Comparison of the spectra for peptide 270–278 from $PPM1D_{\Delta hinge}$ with $PPM1D_{1-420}$ showed an increase in the $t_{1/2}$, the exposure time at which the two populations observed in the spectra are equal, from less than 10 s in wild-type to roughly 10 min in $PPM1D_{\Delta hinge}$. Thus, deletion of the hinge slows the movement and rate of deuterium exchange in the flap.

Mapping the differences in exchange as a result of these deletions onto our structural model reveals that they all colocalize to the same face of the protein core (Fig. 4F). Taken together,

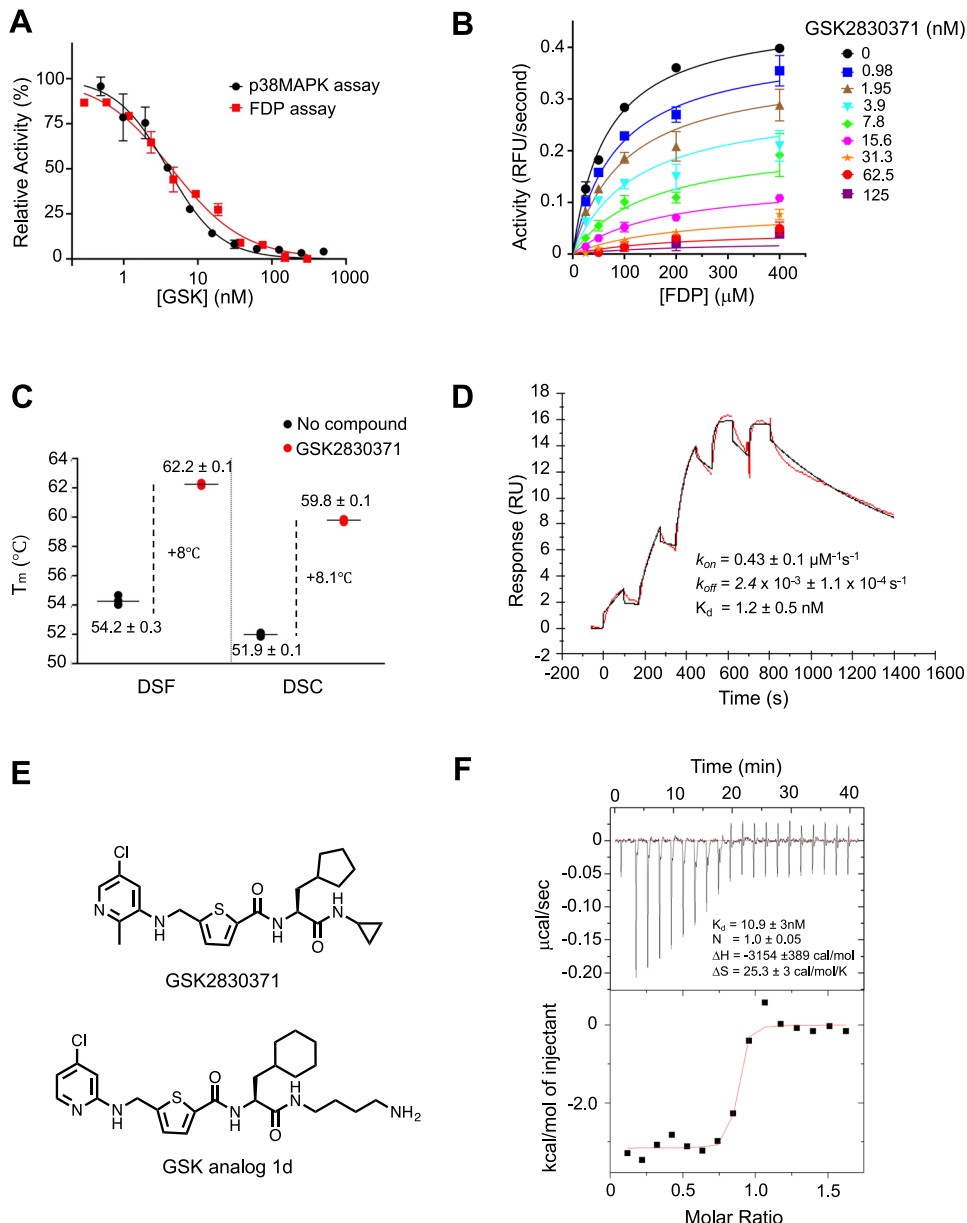

**Fig. 3 GSK2830371 functions as a non-competitive inhibitor and binds with high affinity to PPM1D at an allosteric site. A** Activity of GSK2830371 in the FDP (red) and p38 MAPK (black) enzymatic assays. FDP is cleaved by PPM1D in a substrate-independent fashion while p38 MAPK threonine 180 is dephosphorylated in a substrate-specific manner. $N = 3$ independent samples runs. Data are presented as mean values ± SD. **B** Mechanism of inhibition studies using non-linear fit of Michaelis–Menten enzyme kinetics where hydrolysis of FDP is measured in relative fluorescence units (RFU). Data suggest noncompetitive mode of action with $\alpha \geq 1$. $N = 2$ independent samples runs. Data are presented as mean values ± SD. **C** Differential scanning calorimetry (DSC) and fluorimetry (DSF) showing the melting temperature of $PPM1D_{1-420}$ in the absence (black) and presence (red) of GSK2830371. $N = 3$ independent sample runs. Data are presented as mean values. **D** Surface plasmon resonance analysis of the interaction of GSK2830371 with $PPM1D_{1-420}$. The $K_d$ is calculated from the $k_{on}$ and $k_{off}$ measurements. **E** Structure of GSK2830371 and the GSK2830371 analog 1d used in the isothermal calorimetry experiments. **F** The thermodynamic profile of the interaction of the GSK2830371 analog (1d) with $PPM1D_{1-420}$ as measured by isothermal calorimetry. $N = 3$ independent sample runs. Source data are provided in the Source Data file.

these data indicate that the flap region moves between two conformations and interacts with specific, colocalized regions of the protein. This flexibility appears to be mediated by the hinge, as deletion of residues 155–166 restricts the movement of the flap region and leads to a more stable interaction with the core of the protein.

**Binding of GSK2830371 traps PPM1D in one conformational state.** We hypothesized that movement of the flap region, and thus the conformation of the protein, may be influenced by

binding to GSK2830371. To address this directly, we repeated the HDX-MS experiments in the presence and absence of the compound. Comparison of $PPM1D_{1-420}$ upon addition of GSK2830371 revealed a significant decrease in deuterium uptake in two of the regions exposed upon flap deletion, residues 101–120 and 152–167, suggesting that these regions are either directly involved in GSK2830371 binding or that binding causes a significant structural change in these regions (Fig. 5A). Moreover, GSK2830371 binding resulted in the near loss of bimodal distribution kinetics in the flap region, indicative of a shift of

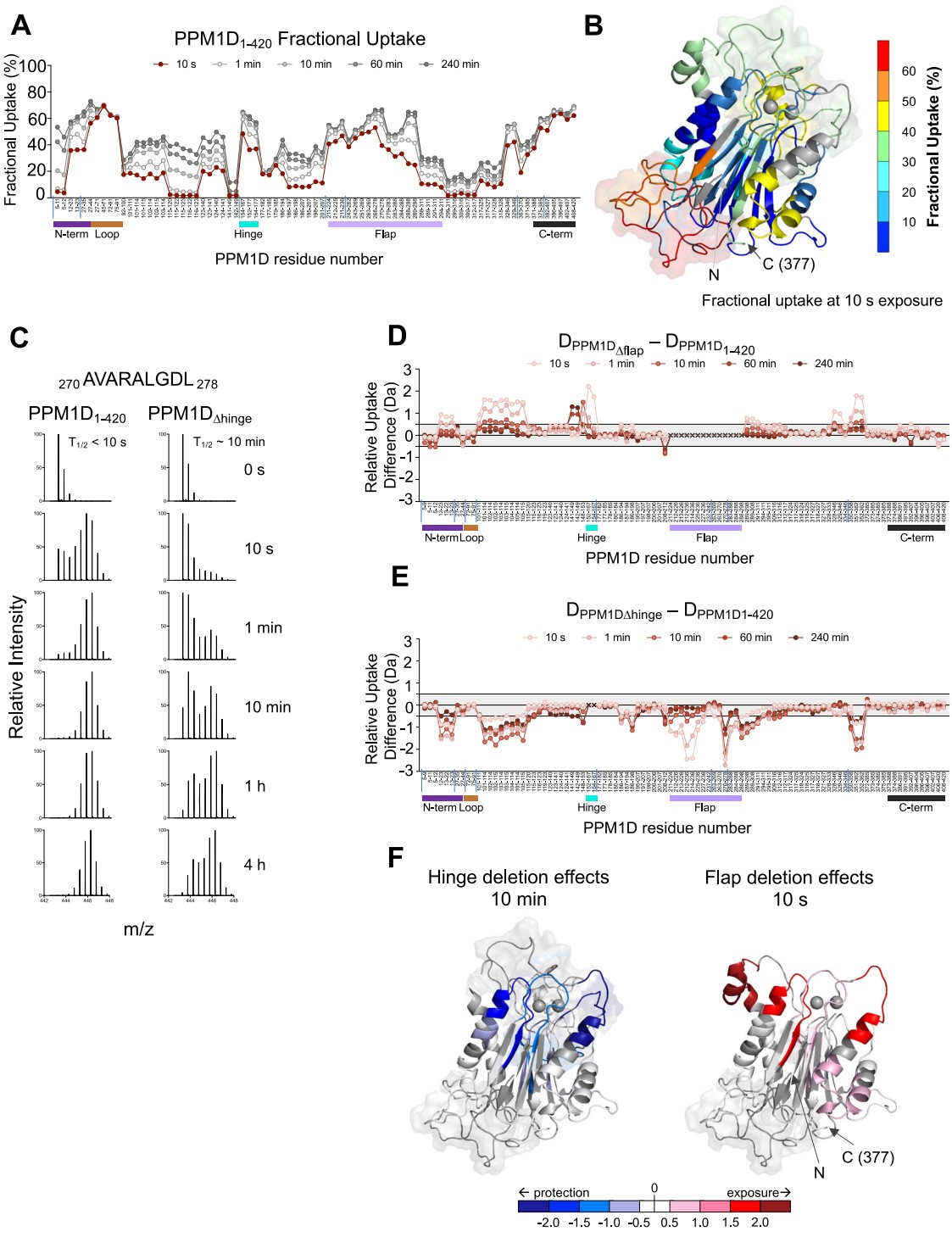

**Fig. 4 PPM1D exists in multiple conformational states reflected by movement of the flap domain. A** HDX-MS measurements of deuterium uptake for PPM1D$_{1-420}$ over time. The peptides analyzed run along the *X*-axis from the N-terminus (left) to C-terminus (right) and vertical tick marks between peptides are shown to indicate regions of missing coverage. Deuterium incorporation is represented as percent fractional uptake on the *Y*-axis, calculated as described in the "Materials and Methods" section. The 10 s exposure time point is shown in red to highlight the uptake values mapped in panel **B**. Reference plots are reported in Suppl. Data 1, the raw values of the plots are reported in Suppl. Data 2 **B** Fractional uptake at 10 s exposure for PPM1D$_{1-420}$ mapped onto the homology model. **C** Spectral analysis showing the change in EX1 kinetics for peptide $_{270}$AVARALGDL$_{278}$ located in the flap region in PPM1D$_{1-420}$ and PPM1D$_{\Delta hinge}$. **D** Relative differences in deuterium uptake between PPM1D$_{\Delta flap}$ compared to PPM1D$_{1-420}$. **E** Relative differences in deuterium uptake between PPM1D$_{\Delta hinge}$ compared to PPM1D$_{1-420}$. **F** Differences shown in panels (**D**, **E**) between PPM1D$_{\Delta hinge}$ (left) or PPM1D$_{\Delta flap}$ (right) and PPM1D$_{1-420}$ mapped onto homology model. Negative differences are represented in blue indicating protection from exchange, while positive differences are shown in red, indicating exposure to exchange as a result of the deletion. Darker colors indicate a stronger relative difference. For panels (**A**, **D**, **E**) the bars below the *X*-axis represent the N-terminus (purple), loop (brown), hinge (cyan), flap (light purple), and C-terminus (black) of the protein. Source data are provided as a Source Data file.

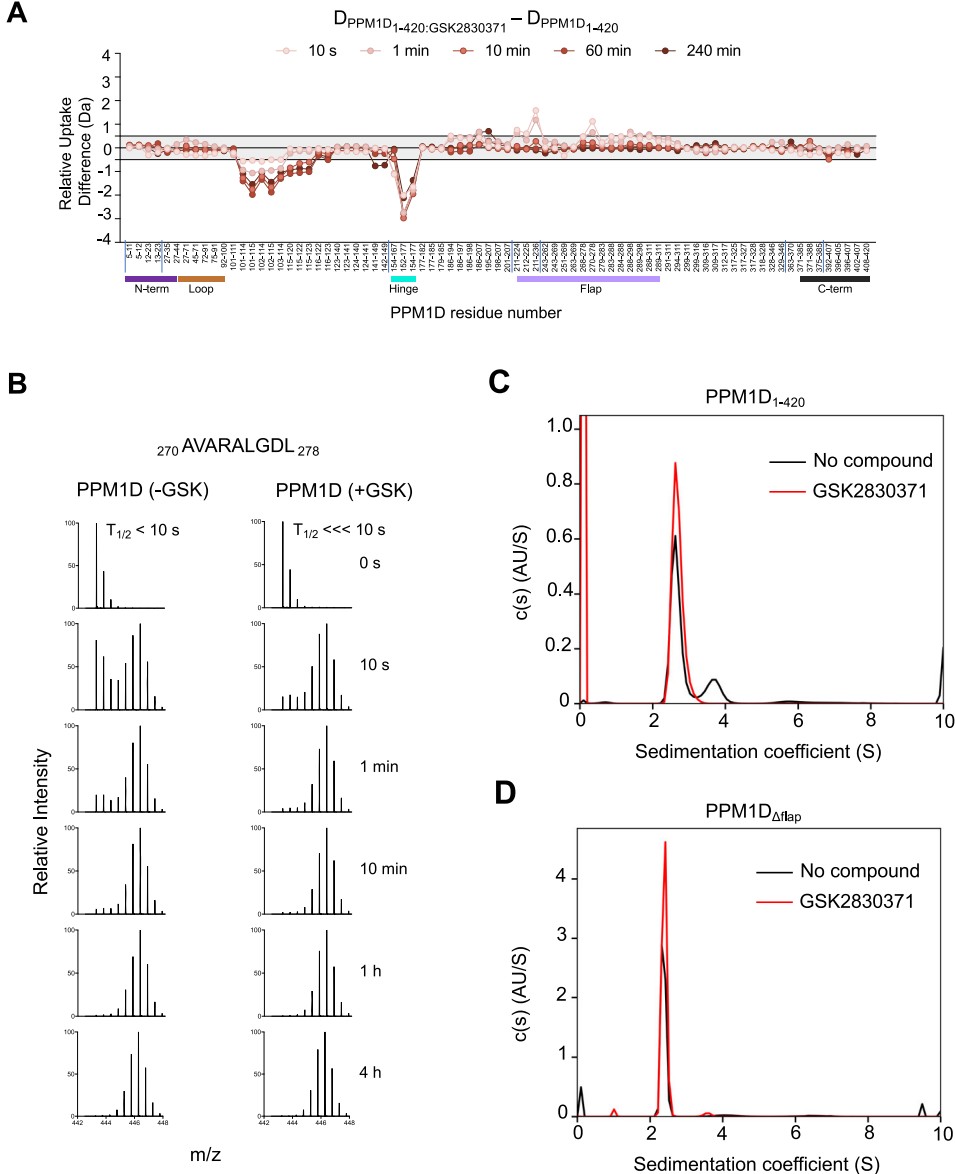

**Fig. 5 GSK2830371 binding alters the conformational sampling of the flap domain of PPM1D. A** Relative differences in deuterium uptake between PPM1D$_{1-420}$ in the presence of GSK2830371 compared to PPM1D$_{1-420}$ without GSK2830371 as assessed by HDX-MS. Vertical tick marks between peptides on the *X*-axis are shown to indicate regions of missing coverage. The bars below the *X*-axis represent the N-terminus (purple), loop (brown), hinge (cyan), flap (light purple), and C-terminus (black) of the protein. Reference plots are reported in Suppl. Data 1, the raw values of the plots are reported in Suppl. Data 2 **B** EX1 analysis showing the change in deuterium uptake for peptide $_{270}$ AVARALGDL $_{278}$ located in the flap region in PPM1D$_{1-420}$ in the absence (left) and presence (right) of GSK2830371. **C** Sedimentation populations of PPM1D$_{1-420}$ in the absence (black) and presence (red) of GSK2830371 as measured by SV-AUC. **D** Sedimentation populations of PPM1D$_{\Delta flap}$ in the absence (black) and presence (red) of GSK2830371 as measured by SV-AUC.

PPM1D from two conformations to a single, flap-exposed conformation (Fig. 5B).

We next sought to confirm the HDX-MS results using an orthogonal approach. To this end, we performed sedimentation velocity analytical ultracentrifugation (SV-AUC) on PPM1D$_{1-420}$ in the absence and presence of GSK2830371[31]. SV-AUC can detect and quantify the sedimentation velocity of different species of a macromolecule which may exist because of differences in shape, mass, or density. In the absence of GSK2830371, PPM1D$_{1-420}$ sedimented into two species at 2.63S (73%) and 3.5S (27%) (Fig. 5C), supporting the presence of two conformations. The observed equilibrium is likely due to a conformational or shape change, rather than the formation of a stable dimer, since PPM1D is well-established to be monomeric in solution[32].

Repeating the experiment in the presence of GSK2830371 yielded a single peak at 2.69S, which supports a model where the compound stabilizes PPM1D into one specific conformation (Fig. 5C). Moreover, the percentage distribution of PPM1D in the two species closely aligns with the bimodal distribution observed in residues 270-278 within the flap region, where ~70% of the protein is in the exposed population in the absence of GSK2830371, while >95% is in the exposed population in the presence of GSK2830371. The HDX data suggested that the existence of two conformational states could be reflected in the movement of the flap region; deletion of the flap region would therefore be predicted to give rise to a single conformational state in the SV-AUC data. Indeed, SV-AUC of PPM1D$_{\Delta flap}$ revealed a single conformation in the absence or presence of GSK2830371

(Fig. 5D). Finally, deletion of the flap region resulted in complete loss of activity in both FDP and P38 MAPK enzymatic assays, further confirming that the single conformation that exists upon deletion of the flap-domain is catalytically inactive (Supplementary Fig. 4B, C).

Taken together, the enzymatic, biophysical, HDX-MS, and SV-AUC data indicate that PPM1D exists in two conformations characterized by the movement of the flap region. Binding of GSK2830371 shifts the equilibrium into a single conformation, yielding a catalytically inactive protein.

**The hinge, not the flap region, is required for high affinity binding between PPM1D and GSK2830371.** The initial description of GSK2830371 utilized photoaffinity labeling and chimeric PPM1A and PPM1K proteins to conclude that GSK2830371 binds the flap region and hypothesized that this interaction explained the selectivity for PPM1D compared to its paralogs[5]. We found that PPM1D with and without deletion of the flap region had no significant difference in the thermal shift, binding affinity, binding kinetics, or deuterium exchange profile caused by GSK2830371, suggesting that the flap region is not required for binding (Fig. 6A, B, Supplementary Fig. 5A, B). However, the HDX studies strongly implicate the hinge, a region also unique to PPM1D, in the binding of GSK2830371 (Fig. 5A). We therefore tested the role of the hinge in GSK2830371 binding and inhibition. Deletion of the hinge caused a ~1000-fold decreased binding affinity to GSK2830371 by SPR and 100-fold higher $IC_{50}$ in the FDP and p38 MAPK assays, clearly implicating its importance in binding of the inhibitor (Fig. 6C, Supplementary Fig. 5C–E).

Though deleting the hinge impaired high-affinity binding of GSK2830371, we compared the deuterium uptake of PPM1D$_{\Delta hinge}$ with and without GSK2830371 and found few changes except for a clear increase in uptake for the flap region (Supplementary Fig. 5F), consistent with the model of compound binding leading to greater conformational freedom of the flap. As discussed previously, deletion of the hinge resulted in dramatic protection throughout PPM1D, presumably due to the change in dynamics of the flap region conformations (Fig. 4D). In the presence of GSK2830371, the flap region appears to be displaced from the core of the protein, but to a much lesser extent than in the wild-type protein. The $t_{1/2}$ is roughly 10 s for the flap region peptide in PPM1D$_{\Delta hinge}$:GSK2830371, while a protected population is undetectable in PPM1D$_{1-420}$:GSK2830371 (Fig. 6D). While these data indicate that some binding does persist in the absence of the hinge, the flap region is still able to interact with the core of the protein, resulting in less efficient inhibition by GSK2830371. These data demonstrate that the hinge, not the flap, is essential for the high-affinity binding and potency of GSK2830371.

**Truncating mutations proximal to residue 400 destabilize the protein and are rare in human disease.** Somatic C-terminal truncating mutations in *PPM1D* almost exclusively cause truncations distal to residue 400 (Fig. 7A). Though known to cause increased abundance of catalytically active PPM1D via impaired proteasome-mediated degradation, the effect of the mutations on protein structure and conformation, are unknown[3]. Consistent with these findings, the saturation mutagenesis of PPM1D$_{1-427}$ revealed enhanced activity with the introduction of stop codons distal, but not proximal to residue 400 (Fig. 2C, Supplementary Fig. 1C, Source Data File). To interrogate this finding further we generated two deletion constructs: PPM1D truncated after residue 400 (PPM1D$_{1-400}$) and PPM1D truncated after residue 377 (PPM1D$_{1-377}$).

While deletions proximal to residue 420 altered the secondary structure of the protein as assessed by circular dichroism, deletion of residues 400–420 had minimal effect on protein stability ($T_m$ difference of −1 °C) and deuterium uptake, whereas deletion of residues 378–420 destabilized the protein ($T_m$ difference of −4.5 °C) (Fig. 7B, C and Supplementary Fig. 6A, B). Moreover, PPM1D$_{1-377}$ exhibited significantly more deuterium uptake in the N-terminal residues 1–35 compared to PPM1D$_{1-400}$ (Fig. 7D). These same N-terminal residues are predicted to be structured in the homology model and required for PPM1D function in the saturation mutagenesis screen and genetic analyses. Furthermore, loss of the N-terminal residues results in insoluble, aggregated recombinant protein which precluded further characterization of the protein lacking the N-terminus. In contrast, the more distal N-terminal residues 39–95 in the loop are not required for stability or activity, and the loss of these residues does not destabilize the protein or lead to exposure of the C-terminus (Supplementary Fig. 6C). These data therefore support the existence of a stabilizing interaction between the N-terminus and C-terminal residues 378–400, consistent with predictions from the homology model that the N- and C- termini are in close proximity and form interacting β-sheets (Fig. 1A). Thus, the destabilizing effects of truncating mutations proximal to residue 400 of *PPM1D* likely explain the rarity of these mutations in human disease. In contrast, mutations distal to residue 400 which retain protein stability are common, as they impair proteasomal degradation, thereby conferring chemoresistance via suppression of the DDR.

## Discussion

Our findings demonstrate that PPM1D exists in two protein conformations, characterized by the movement of the flap region, and that GSK2830371 locks PPM1D into a single, inactive conformation, providing a mechanistic basis for allosteric enzymatic inhibition. Furthermore, high-affinity, PPM1D-selective binding of GSK2830371 is mediated by the hinge, which is unique to PPM1D, providing a basis for the striking selectivity of this molecule for PPM1D among related phosphatases. These results were confirmed with orthogonal biophysical, biochemical, and structural assays and contextualized using a homology model that made predictions consistent with computational and functional genetic observations.

At baseline, PPM1D exists in two conformations. Our HDX-MS data revealed bimodal distributions in the flap region, suggesting that movement of the flap region dictates the equilibrium between these conformational states and that this flap movement is a response to the position and flexibility of the hinge. We confirmed the existence of these two states, and their relative abundance predicted by HDX-MS, using SV-AUC, an orthogonal experimental method. Deletion of the flap region caused a loss of the second conformation by SV-AUC, again confirming the importance of the flap region in mediating the conformational equilibrium. The addition of GSK2830371 caused a shift to a single conformation in both HDX-MS and SV-AUC assays.

The conformational changes induced by binding of GSK2830371 may be due to a structural change as a result of binding to a single site. Alternatively, the drug may bind to multiple sites of the protein, serving as an intra-molecular "glue," akin to what was described for the SHP2 inhibitor, SHP099[33]. The multiple protected regions observed after GSK2830371 could be consistent with direct drug interaction but also could be a result of indirect structural changes. Further efforts to define the site or sites of drug binding may inform these findings.

Whereas a prior report describing GSK2830371 suggested that the allosteric binding site resides in the flap region[5], we found

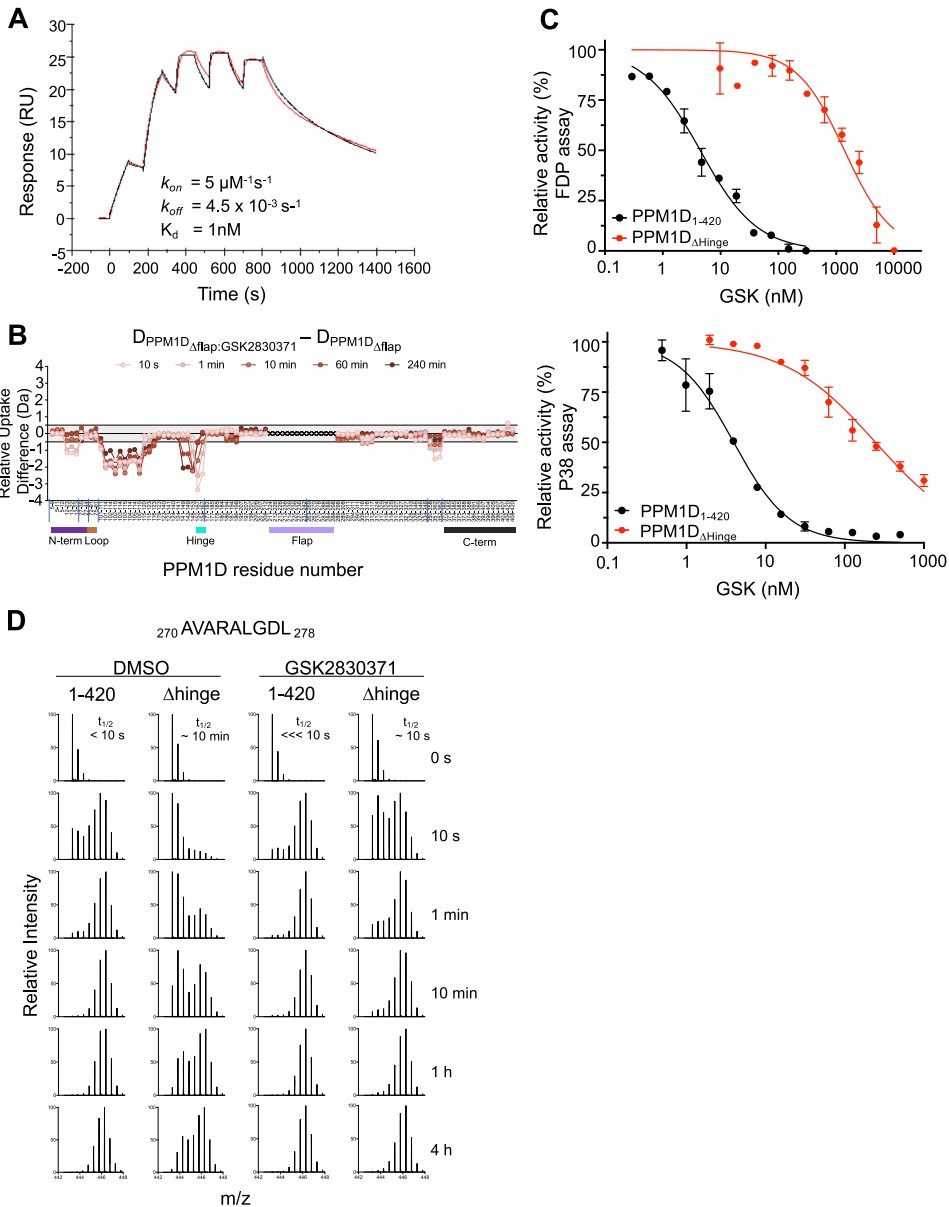

**Fig. 6 The Hinge, not the flap, mediates GSK2830371 binding. A** Surface plasmon resonance of PPM1D$_{\Delta flap}$ and GSK2830371 (red represents raw data and black represents fitted kinetic parameters). **B** Relative differences in deuterium uptake between PPM1D$_{\Delta flap}$ in the presence of GSK2830371 compared to PPM1D$_{\Delta flap}$ in the absence of GSK2830371 as assessed by HDX-MS. Vertical tick marks between peptides on the X-axis are shown to indicate regions of missing coverage. The bars below the X-axis represent the N-terminus (purple), loop (brown), hinge (cyan), flap (light purple), and C-terminus (black) of the protein. **C** Activity of GSK2830371 in the FDP (top) and p38 MAPK (bottom) enzymatic assays with PPM1D$_{1-420}$ (black) or PPM1D$_{\Delta hinge}$ (red). $N = 2$ independent samples runs. Data are presented as mean values ± SD. **D** EX1 analysis showing the change in deuterium uptake for peptide $_{270}$AVARALGDL$_{278}$ located in the flap region in PPM1D$_{1-420}$ and PPM1D$_{\Delta hinge}$ in the absence (left) and presence (right) of GSK2830731. Source data are provided in the Source Data file.

that loss of the flap region had no effect on the binding of GSK2830371 to PPM1D. Instead, we identified the hinge, residues 155–166, as the key mediator of this high-affinity interaction. The selectivity of GSK2830371 for PPM1D suggested that it binds a site unique to and important for PPM1D function, a requirement that only the hinge fulfills. Furthermore, GSK2830371 binding strongly protected the hinge from deuterium exchange suggesting a role in the interaction. Deletion of the hinge decreased the binding affinity of GSK2830371 to PPM1D by approximately 1000-fold, and GSK2830371 was nearly 100-fold less active in enzymatic assays in the absence of the hinge. Despite this, some activity and HDX modulation was observed in the absence of the hinge, which may be attributed to residues 100–120, where

GSK2830371 binding led to substantial protection in the wild-type protein. This region likely also plays a role in compound binding, though the functional and potential structural importance of these residues by saturation mutagenesis would complicate the interpretation of residues 100-120 deletion studies.

Our and other groups' inability to crystalize the protein is likely a result of the presence of the dynamic regions of the protein that prevent the protein molecules from efficiently packing during the vapor diffusion process. Instead, we utilized a homology model to gain insights into the protein structure. The model made several predictions that were supported by the computational genetic data, the saturation mutagenesis studies, and the HDX-MS data. For example, in contrast to structured residues predicted to be

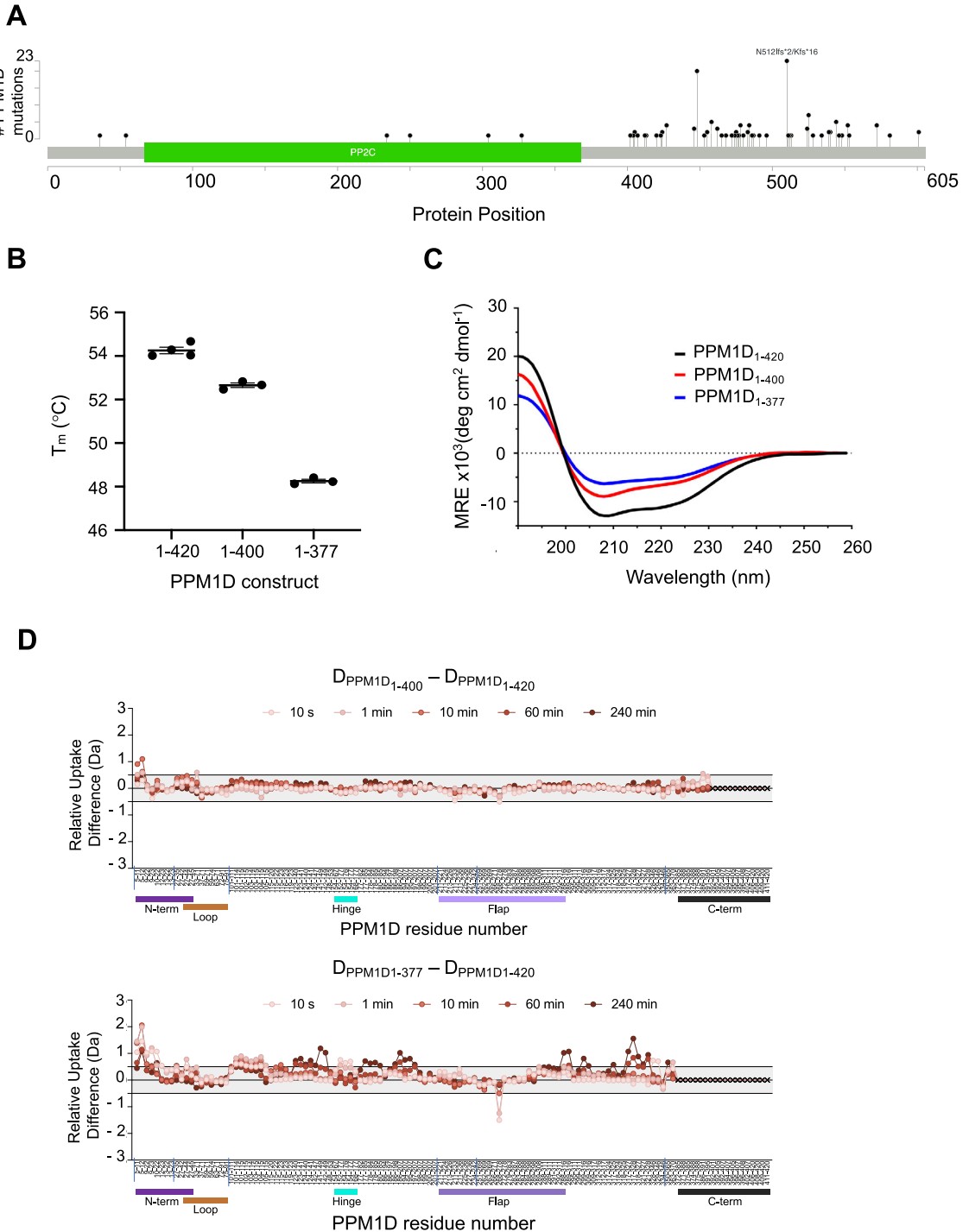

**Fig. 7 Truncations proximal to amino acid 400 destabilize PPM1D. A** Distribution of PPM1D truncating mutations in solid tumor sequencing data from GENIE (v5.0) obtained from cBioportal (www.cbioportal.org). **B** Effects of different truncations on the thermal stability of PPM1D by differential scanning calorimetry. $N = 3$ independent sample runs. Data are presented as mean values ± SD. Source data are provided as a Source Data file. **C** Circular dichroism analysis of $PPM1D_{1-420}$ (black), $PPM1D_{1-400}$ (pink) and $PPM1D_{1-377}$ (brown). **D** Relative differences in deuterium uptake between $PPM1D_{1-420}$ and $PPM1D_{1-400}$ (top) and between $PPM1D_{1-420}$ and $PPM1D_{1-377}$ (bottom) as assessed by HDX-MS. Vertical tick marks between peptides on the $X$-axis are shown to indicate regions of missing coverage. The bars below the $X$-axis represent the N-terminus (purple), loop (brown), hinge (cyan), flap (light purple), and C-terminus (black) of the protein.

inward facing and part of the core catalytic domain, the predicted unstructured domain of residues 39–95 were not evolutionarily conserved, genetically constrained, or functionally important in the suppression of the DNA damage response, and had no impact on the deuterium exchange throughout the rest of the protein.

Our data also provide insight into the impact of C-terminal truncation mutations in cancer. Mutations proximal to residue 400 are rare in human malignancies, and our saturation mutagenesis study confirmed that, in contrast to truncations distal to 400, those proximal to 400 impaired activity. Furthermore, truncation at residue 378 lead to significantly greater

destabilization of the protein and exposure of N-terminal residues compared to truncation at 400. Therefore, while previous work has shown that C-terminal truncations of PPM1D result in elevated levels and activity caused by impaired protein degradation[3], we found that more proximal truncations impair protein function and stability.

The therapeutic potential of selectively inhibiting PPM1D spans across tissue types and therapeutic contexts. A selective and potent allosteric inhibitor with acceptable pharmacokinetic properties could be applied to tumors with *PPM1D* alterations and combined with DNA damaging modalities to augment anti-tumor activity. As shown for PPM1D and GSK2830371, understanding the basis of selectivity, mode of activity, and mechanism of inhibition can empower rational drug design and discovery efforts.

## Methods

**Homology model generation**. Using BLAST, we obtained the structures of human PPM1A (PDB ID: 3FXJ, https://www.wwpdb.org/pdb?id=pdb_3FXJ), human PPM1B (PDB ID: 2P8E, https://www.wwpdb.org/pdb?id=pdb_2P8E), and a PPM-type phosphatase (PDB ID: 2I0O, https://www.wwpdb.org/pdb?id=pdb_2I0O). Any additional information can be requested from the corresponding author. The sequences were aligned with constraints on the 21MED23 region that are required for the catalytic function of PPM1D, and knowledge-based alignments were made with Clustal Omega[17,19]. The aforementioned structures were then used as templates for multi-sequence modeling followed by loop refinement using the software package Modeller[18]. Since the metal-binding sites (D60, G61, D239, D282) were predicted to be surrounded by several loop regions, the sidechains and backbones of the corresponding residues were re-oriented and locally optimized using two $Mg^{2+}$ ions in reference to the aligned metal-binding sites of PPM1A. The entire structure was relaxed with the AMBER ff14SB force field.

**Saturation mutagenesis**. The principle of saturation mutagenesis has been previously described by our group and others[24,25]. The PPM1D library contains 16,377 cDNAs encoding for all possible amino acid changes in a PPM1D from amino acids 1–427. The generation of oligonucleotides pools cloning into the lentiviral backbone pMT_BRD025 which allows for ORF expression under control of the human EF1α promoter, and production of lentivirus has been previously described[25]. The generation of K562-TP53wild-type -CDKN1A-GFP reporter cells has been previously described[24]. Briefly, the endogenous CDKN1A gene in K562-TP53wild-type cells was tagged at the carboxy terminus with GFP using the eFlut (endogenous Fluorescent tagging) methodology as previously described[34]. The entire screen was performed in duplicate. After two independent rounds of infection of the K562-TP53wild-type—CDKN1A-GFP cells at a low multiplicity of infection, cells were selected with 2 ug/ml puromycin (ThermoFisher, #A1113802) for two days. Two days after removal of puromycin selection, infected cells were treated with 200 nM of daunorubicin and 24 h later GFPhi and GFPlo cells from either treatment arm were collected by flow-sorting and frozen as cell pellet at −80 °C (2 × 10^6 − 10 × 10^6 cells per sample). FACS was done on a BD FACSAria II using FACSDiva v8 and analyzed on FlowJo v10.

Genomic DNA was isolated (QIAamp DNA Blood Midi kits, QIAGEN, #51183), and ~1.5 µg of each sample was subject to 24 PCR reactions using Q5 High-Fidelity DNA polymerase (New England Biolab, # M0491L) to amplify the ORFs (PCR forward primer: 5′ ATTCTCCTTGGAATTTGCCCTT 3′, reverse primer: 5′ CATAGCGTAAAAGGAGCAACA 3; PCR cycling program: 98C 30 s; 35 cycles of [98C 10 s; 64C 30 s; 72C 2 min]; 72C 2 min; 4C hold). The PCR reactions for each sample were pooled and concentrated (QIAquick PCR Purification Kit,QIAGEN, #28104), then separated by gel electrophoresis (1% agarose). DNA was purified from excised bands (QIAquick Gel Extraction Kit, QIAGEN, #28704; AMPure XP beads, Beckman Coulter, #A63880) and prepared per the Illumina Nextera XT protocol. For each sample, six Nextera reactions, each with 1 ng of purified ORF DNA, were setup and indexed with unique i7/i5 index pairs. After the limited cycle PCR step, the 6 Nextera reactions combined, purified (AMPure XP beads,Beckman Coulter, #A63880), quantified, pooled by equal weight, and detected using the NovaSeq 600 Sequencing System (Illumina) using two reads, each 150 bases in length, and 2 index reads of 8 bases long. Sequencing data were processed with the ORFcall software developed by the Broad Institute which aligns read pairs to the PPM1D reference sequence, allowing for quantification of each variant[25]. 64 possible codons are possible at each codon position, which includes both variants built into the library and those that result from errors in both library synthesis and PCR/sequencing. Quality control to distinguish intended from unintended variants were performed as described[25]. After removal of unintended variants, the raw read counts were normalized to the representation of reads at each codon position. We then calculated the log10 ratios of fractional read counts in GFPlo over GFPhi cell populations for all non-synonymous and synonymous PPM1D variants. These enrichment scores

expressed as $log_{10}(GFP^{lo}/GFP^{hi})$ were depicted as heatmaps and aligned to the other analyses described in the manuscript. Image 2A was created with Biorender.com.

**Protein expression and purification**. The gene encoding full length PPM1D was codon-optimized for bacterial expression and synthesized commercially (GeneArt, Life Technologies). PPM1D$_{1–420}$ was PCR-amplified from the synthesized gene with BsaI sites and cloned using Golden Gate cloning into a derivative of pET21b that was designed to append an N-terminal fusion tag comprised of a His(6) tag and a SUMO protein. PPM1D$_{1–420, Δ39–95}$ (PPM1D$_{Δloop}$), PPM1D$_{1–218-GGSNGS-288–420}$ (PPM1D$_{Δflap}$), PPM1D$_{1–400}$, PPM1D$_{1–377}$ and PPM1D$_{1–154-GGS-167–420}$ (PPM1D$_{Δhinge}$) were produced by site-directed mutagenesis of the PPM1D$_{1–420}$ vector using the Q5 SDM kit (New England Biolabs, Ipswich, MA). Similarly, PPM1D$_{1–420}$, PPM1D$_{Δflap}$ and PPM1D$_{Δhinge}$ were each inserted into pET21b with N-terminal His(8)−28mer-His(8) for the SPR experiments. Correct insertion of the genes into the expression vectors was confirmed by Sanger sequencing. The proteins were expressed in BL21 (DE3) cells (New England Biolabs, Ipswich, MA). The cells were grown at 37 °C before inducing with 0.5 mM IPTG and 2 mM $MgCl_2$ and left to grow overnight at 18 °C. The cells were pelleted by centrifugation and stored at −80 °C until use.

Cell pellets were resuspended in 50 mM Tris-HCl pH 7.4, 500 mM NaCl, 10% glycerol, 10 mM MgCl2, 1 mM TCEP, 1X EDTA-free cOmplete protease inhibitors (Roche) and 1/1000 volume of Lysonase (EMD-Millipore) and lysed using a microfluidizer (Avestin, Canada). The soluble protein was extracted by centrifugation at 15,000 rpm and filtered before loading onto 2 × 5 mL stacked Ni$^{2+}$-charged HisTrap columns (GE Healthcare). 10 mM Imidazole was added to the lysate before loading onto the column. Running buffers for the column were as follows: Buffer A contained 50 mM Tris-HCl pH 7.4, 500 mM NaCl, 10% glycerol,10 mM $MgCl_2$, 1 mM TCEP and Buffer B contained an additional 400 mM imidazole. The column was washed with 10% Buffer B before applying a gradient of 10–100% Buffer B over 20 column volumes. Combined fractions were dialyzed against 2 L buffer containing 50 mM Tris-HCl pH 7.4, 500 mM NaCl, 10% glycerol,10 mM MgCl2, 1 mM TCEP with 1:1000 (wt/wt) of His6-Ulp1 protease to cleave the His6-(SUMO) tag. The His(6)-SUMO and His(6)-Ulp1 protease was removed by passing the solution over the HisTrap column a second time. The flow through of the HisTrap column was concentrated and passed over a Superdex 200 26/600 column equilibrated in 50 mM Tris-HCl pH 7.4, 500 mM NaCl, 10% glycerol,10 mM MgCl2, 1 mM TCEP. The proteins were concentrated and stored at −80 °C until use.

The gene encoding the p38 MAPK with the stabilizing mutation C162S and the constitutively active version of MKK6(S207E, T211E) was codon-optimized for bacterial expression and synthesized commercially (GeneArt, Life Technologies). p38 MAPK$_{2–360, C162S}$ and MKK6$_{2–334, S207E, T211E}$ were PCR-amplified from the synthesized genes with BsaI sites and cloned using Golden Gate cloning into a derivative of pET21b that appends an N-terminal His(6) tag. Correct insertion of the genes into the expression vectors was confirmed by Sanger sequencing. The proteins were expressed as described above, but without the addition of $MgCl_2$. Cells were pelleted by centrifugation and stored at −80 °C until use. Cell pellets were resuspended in 50 mM Tris pH 8, 500 mM NaCl, 20 mM Imidazole, 1 $MgCl_2$, 100 µM TCEP, 1X EDTA-free cOmplete protease inhibitors and 1/1000th vol of lysonase, and lysed using a microfluidizer. The solution was clarified by centrifugation at 10,000 rpm for 25 min and filtered before loading onto 2 × 5 mL stacked Ni$^{2+}$-charged HisTrap columns (GE Healthcare). Buffer A contained 50 mM Tris-HCl pH 8.0, 500 mM NaCl, 1 mM $MgCl_2$, 100 µM TCEP and Buffer B contained an additional 1 M imidazole. The column was washed with 2% Buffer B before applying a gradient of 2–52% buffer B over 10 column volumes. The peak fractions from the Histrap were pooled, TCEP added to give a final concentration of 0.5 mM, and the protein passed over a Superdex 200 26/600 column that was equilibrated with 50 mM Tris pH 8, 500 mM NaCl, 100 µM TCEP. At this point, MKK6 was dialyzed into 20 mM HEPES pH 7.4, 150 mM NaCl, 10% glycerol, 5 mM $MgCl_2$, 500 µM TCEP, concentrated, and stored at −80 °C, with the His(6) tag uncleaved for downstream applications. The peak fractions from the Superdex runs of p38 MAPK were combined and concentrated and TEV protease added at a ratio of 1:50 TEV protease:p38 MAPK (w/w). After incubation overnight at 4 °C, imidazole was added to give a final concentration of 20 mM and the solution passed over a 5 mL HisTrap column to remove the affinity tag and the His-tagged TEV protease. The cleaved protein was dialyzed overnight into 20 mM HEPES pH 7.4, 150 mM NaCl, 10% glycerol, 5 mM $MgCl_2$, 500 µM TCEP, concentrated, and stored at −80 °C until use. The yields from both the purifications were in excess of 60 mg/L of TB culture growth.

**Intact mass analysis**. Purified proteins were diluted to 100 ng/µl in 25 mM HEPES pH 7.5, 10 mM $MgCl_2$, 150 mM NaCl. 1 µl diluted protein was injected onto a Waters BioAccord LC-TOF (composed of an ACQUITY I-Class UPLC and RDa detector with an ESI source) outfitted with an ACQUITY UPLC Protein BEH C4 column (300 Å, 1.7 µm, 2.1 × 50 mm) held at 80 °C for the duration of analysis. Samples were desalted for 1 min before being eluted onto the MS with a 5% to 85% gradient of acetonitrile in 2.5 min at a flow rate of 0.4 ml/min. Ionization was performed at a cone voltage of 55 V and desolvation temperature of 550 °C. The

instrument scanned at a rate of 0.2 scans/s over the range 50–2000 $m/z$. Protein $m/z$ spectra were deconvoluted into intact mass using the MaxEnt1 function within UNIFI software (Waters, Inc.).

**Differential scanning fluorimetry (DSF) and differential scanning calorimetry (DSC) measurements**. The purified protein constructs were mixed in the assay buffer containing 25 mM HEPES pH 7.5, 10 mM MgCl$_2$, 150 mM NaCl. The sample preparation for the DSF assay included incubating the protein constructs (final concentration 0.2–0.4 mg/mL) with either DMSO alone (control) or with 40 μM GSK2830371 for 30 min before the addition of the Sypro orange dye (final concentration 10×). The final reaction volume of 10 μL was pipetted into a 384 multi-well clear plate (Light Cycler 480), sealed using a foil and spun down at 1000 × $g$ for 2 min. The fluorescence emission intensity of the Sypro Orange dye was monitored in a LightCycler-480 instrument by running the Protein Melt program. The program was run in a continuous acquisition mode from 25 °C to 95 °C with a ramp rate of 0.06 °C/s (with 10 acquisitions/°C). The data (first derivative of the fluorescence signal as a function of temperature) was analyzed using the LightCycler-480 software for measuring the $T_m$ values with and without the compound. A separate control with buffer alone and buffer with compound was run in the same plate to subtract out the potential interactions of the buffer components and/or the compound with the dye.

DSC thermograms were recorded in MicroCal DSC from Malvern with an autosampler. For DSC experiments the purified protein constructs were extensively dialyzed against 25 mM HEPES pH 7.5, 10 mM MgCl$_2$, 150 mM NaCl and 1 mM TCEP. The experiments were performed using 0.4 mg/mL concentration of wild-type or mutant protein (8.5–8.9 μM), with and without GSK2830371 compound at 40 μM and 2% DMSO (v/v). The respective reference scans were run under identical DSC set up conditions and subtracted from each sample scan. The program was run in a continuous acquisition mode from 25 °C to 90 °C at a rate of 120 °C/h The heat capacity curves and midpoint temperature ($T_m$) were analyzed using Origin 7.0 software.

**Isothermal titration calorimetry (ITC) analysis**. Assessments of the analog of GSK2830371 (analog 1d) binding to wild-type and mutant proteins were conducted using an Auto-iTC200 microcalorimeter (Malvern) (see section on synthesis of analog 1d below). The protein was dialyzed extensively into 25 mM HEPES (pH 7.4), 150 mM NaCl, 10 mM MgCl$_2$ and 1 mM TCEP. The dialyzed protein was used to prepare fresh working solutions of protein (20 μM) and analog 1d (200 μM) in 2% DMSO (v/v) in the same buffer. Titrations were performed at 25 °C and consisted of a single initial injection of 0.4 μL, followed by 19 injections of 2 μL of compound into a sample cell containing wild-type or mutant PPM1D, with an interval of 150 s between successive injections. The reference power and stirring speed were set at 10 mCal/s and 1000 RPM, respectively. Heat of dilution and mixing were obtained from injections of analog 1d into ITC buffer lacking protein and subtracted from the integrated data before curve fitting. Thermodynamic data were analyzed using Origin 7.0 software and fitted by non-linear regression to the model for a single binding site.

**Surface Plasmon Resonance (SPR) Analysis**. SPR was carried out on the Biacore S200 (GE Healthcare) using Series S sensor chip NTA (GE BR-1005-32). The preparation of the chip surface was conducted according to the manufacturer's indications. The running buffer contained 50 mM Tris (pH 7.5), 50 mM NaCl, 1 mM MgCl$_2$, 0.5 mM TCEP, 0.005% (v/v) P20 surfactant, and 2% DMSO. PPM1D$_{1-420}$ and PPM1D$_{ΔFlap}$ were run under single-cycle mode due to the slow-off-rate kinetics. The two proteins were immobilized on the chip surface in flow cells 2 and 3 with a signal of ~1500–2000 RU at a rate of 5 μL/min while flow cell 1 was left blank as a reference. The running buffer was injected three times followed by the GSK2830371 analyte in solution. Injections were carried out at a flow rate of 50 μL/min with the analyte in a concentration series of 5-point, 3-fold ending at 400 nM top dose. The association and dissociation phases were monitored for 90 s and 600 s, respectively. Collected data were evaluated using the Biacore S200 Evaluation Software under 1:1 binding mode. PPM1D$_{ΔHinge}$ was run under multi-cycle mode due to its fast-on/fast-off kinetics. The protein was immobilized to give a signal of ~ 3000 RU. GSK2830371 analyte in solution was injected at a flow rate of 90 μL/min in a 10-pt, 2-fold concentration series ending at 15 μM top dose. The association and dissociation phases were monitored for 45 s and 150 s, respectively. Since GSK2830371 had issues with solubility/aggregation at high concentrations that caused artifacts in sensorgrams, only steady-state analysis was performed to extract $K_d$.

**FDP assay**. The FDP assay was adapted from reported protocol with minor modifications[5]. Fluorescein diphosphate tetra-ammonium salt (CAS 217305-49-2) was purchased from AAT Bioquest (Cat No.11600). The assay was carried out on a 384-well plate (Corning Cat No.3575) with assay buffer containing 50 mM Tris (pH 7.5), 50 mM NaCl, 30 mM MgCl$_2$, 0.01% tween 20, and 2% DMSO. Reactions were initiated with the addition of FDP to a solution containing wild-type PPM1D or one of its variants, to a final concentration of 5 nM for PPM1D and 100 μM for FDP (or as indicated in $K_{cat}/K_M$ measurements). A fluorescent signal was detected on a Spectramax microplate reader (485/530 nm) for the next 15 min at room

temperature with an interval for data collection of 45 s. For the IC$_{50}$ measurement, the compound was pre-incubated with PPM1D for 1 h before addition of the substrate.

**p38 MAPK enzymatic assay**. To generate the double phosphorylated form of p38 MAPK for the LC/MS enzymatic PPM1D assay, 20 μM p38 MAPK was incubated with 100 nM MKK6 in 50 mM HEPES pH 7.4, 10 mM ATP, 40 mM MgCl$_2$, and 100 μM TCEP at RT for 2 h. The reaction mixture was diluted 2-fold with 20 mM HEPES pH 7.4, 300 mM NaCl, 100 μM TCEP, and 1 mM MgCl$_2$ and passed over a Superdex 200 26/600 column equilibrated with 20 mM HEPES pH 7.4, 150 mM NaCl, 100 μM TCEP, and 1 mM MgCl$_2$ to remove excess ATP and MKK6 from the phosphorylated p38 MAPK (p38PP) protein. The protein was concentrated and stored in aliquots for single time usage at −80 ˚C.

The enzymatic activity of the PPM1D constructs was measured by quantifying the dephosphorylation of the substrate protein p38PP by LC/MS. Assays were run in 384-well plates (Axygen; #P-384-120SQ-C) in a final volume of 80 μL per well. The buffer consisted of 20 mM HEPES, 20 mM NaCl, 20 mM MgCl$_2$ and 0.1 mM TCEP, at a pH of 7.5. The reaction was initiated by addition of the p38PP substrate to a solution containing 10 nM PPM1D to a final concentration of 300 nM for p38PP. The plate was then incubated at room temperature and the reaction quenched after 15 min increments with formic acid to a final concentration of 6% (v/v) up to 120 min. For IC$_{50}$ measurements, the GSK2830371 compound was pre-incubated with the PPM1D constructs in a 2% DMSO-containing buffer for 10 min prior to the addition of the substrate and the reactions were quenched after 2 h. In both experiments, BSA was added to a final concentration of 0.2 μM during the quench step as a carrier protein and injection control for the LC/MS. The plate was then sealed and spun down at 1500 × $g$ for 1 min prior to LC/MS analysis. Plates were analyzed within the same day of the quench step.

Substrate and product detection was achieved by multiple-reaction monitoring (MRM) using a Waters Acquity UPLC chromatography system connected to an Xevo TQ-XS mass spectrometer (Waters, Milford, MA). Source parameters were set as follows: polarity ES+, capillary 2.0 kV, cone voltage, 50 V; collision energy, 5 V; cone gas flow, 200 L/h; collision gas flow, 0.16 mL/min; source temperature, 150 °C; desolvation temperature, 500 °C; and desolvation gas flow, 800 L/h. Analytes were separated using an Agilent PLRP-S Reversed-Phase column (5 μm; 2.1 × 50 mm, 300 Å Agilent Technologies, Santa Clara, CA) at 60 °C. Sample storage temperature was set to 15 °C. Mobile phases consisted of 0.1% (v/v) formic acid (Millipore EMSURE) in LC/MS grade water (JTBaker) and 0.1% v/v formic acid (Honeywell). Initial column conditions were set to 5% water/95% acetonitrile. Proteins were eluted using a 1.5 min gradient of 5–98% acetonitrile at 0.4 ml/min. MassLynx and TargetLynx software (Waters, version 4.2) were used for sample acquisition and data analysis, respectively. Analytes were quantified by integration of peak areas under the curve using the transitions 941.10 to 941.05 $m/z$ for P38PP (substrate) and 939.10 to 939.09 $m/z$ for P38P (product). Areas were normalized for instrumental variation by dividing by the BSA peak area (1209.23 to 1209.2 $m/z$ transition) measured in the same well.

**Synthesis of analog 1d**. For a schematic of the reactions and structures of all compounds see Supplementary Fig. 3D.

*Synthesis of compound 2*. To a mixture of compound 1 (823 mg, 6.40 mmol) and compound 1_1 (1.00 g, 6.40 mmol) in toluene (20.0 mL) and MeOH (1.00 mL) at 25 °C under vacuum, and the reaction mixture was heated to 110 °C for 1 h. The mixture was concentrated, and the residue was added to MeOH (10.0 mL), NaBH3CN (604 mg, 9.61 mmol) was added, and the mixture was stirred at 25 °C for 12 h. LCMS (product RT = 1.093) showed that the reaction was completed. The reaction was quenched by aq. HCl (1 M) until pH = 6 and concentrated to give a residue. The residue was added to water (20.0 mL), and the mixture was filtered, the filter cake was washed with water (10.0 mL × 2), dried under vacuum to give compound 2 (700 mg, 2.08 mmol, 32.5% yield, 80.0% purity) as a yellow solid which was confirmed by 1H NMR (see Supplementary Notes: Spectroscopic Data for Generation of Analog 1d).

*Synthesis of compound 3*. To a mixture of compound 2 (500 mg, 1.86 mmol, 1 eq) and compound 2_1 (345 mg, 1.55 mmol, HCl) in DMF (10.0 mL) was added NMM (470 mg, 4.65 mmol) and HBTU (705 mg, 1.86 mmol) in one portion at 25 °C under N2, the mixture was stirred at 25 °C for 2 h. TLC (Petroleum ether/EtOAc = 3/1, Rf = 0.43) and LMCS showed that the reaction was completed. The mixture was poured into cold water (20.0 mL), and the aqueous phase was extracted with EtOAc (50.0 mL × 2), the combined organic phase was washed with brine (50.0 mL x 2), dried with anhydrous Na2SO4, filtered and concentrated in vacuum. The residue was purified by silica gel chromatography (SiO2, Petroleum ether/EtOAc = 1/0 to 2/1) to give compound 3 (700 mg, 1.61 mmol, 86.3% yield) as a colorless oil which was confirmed by 1H NMR (see Supplementary Notes: Spectroscopic Data for Generation of Analog 1d).

*Synthesis of compound 4*. To a mixture of compound 3 (600 mg, 1.38 mmol) in MeOH (5.00 mL) was added NaOH (6 M, 275 uL) at 0 °C under N2, the mixture

was stirred at 25 °C for 24 h. TLC (Petroleum ether/EtOAc = 0/1, Rf = 0.01) showed that the reaction was completed. The mixture was acidized to pH = 4 by aq. HCl (1 M), and the mixture was concentrated to give a residue. The residue was added to water (10 mL), and the mixture was extracted with EtOAc (20.0 mL x 3), the combined organic phase was washed with brine (20.0 mL x 3), dried with anhydrous Na2SO4, filtered and concentrated in vacuum to give compound 4 (900 mg, crude) as a white solid which was confirmed by 1H NMR (see Supplementary Notes: Spectroscopic Data for Generation of Analog 1d).

*Synthesis of Compound 1d.* To a mixture of compound 4 (150 mg, 356 umol) in DMF (5.00 mL) was added NMM (89.9 mg, 889 umol), then HBTU (148 mg, 391 umol) was added to the mixture and stirred for 5 mins at 0 °C under N2, compound 4_4 (73.6 mg, 391 umol) was added to the reaction, the mixture was stirred at 25 °C for 12 h. TLC (Petroleum ether/EtOAc = 1/1, Rf = 0.5) showed that the reaction was completed. The residue was poured into water (50.0 mL) and stirred for 10 mins, the aqueous phase was extracted with EtOAc (30.0 mL × 4), the combined organic phase was washed with brine (30.0 mL × 3), dried with anhydrous Na2SO4, filtered and concentrated in vacuum to obtain a crude compound 5d (100 mg, crude) as a yellow oil.

To a mixture of compound 5d (100 mg, 1.00 umol) in DCM (5.00 mL) at 0 °C under N2, then TFA (222 mg, 1.95 mmol) was added in the reaction, the mixture was stirred at 0 °C for 2 hrs, then heated to 25 °C and stirred for 12 h. TLC (Petroleum ether/EtOAc = 1/1, Rf = 0.1) showed that the reaction was completed. The reaction mixture was concentrated under reduced pressure to remove solvent and saturated aq. NaHCO3 was added drop wised at 0–5 °C until the pH = 8. Then MeOH (3.00 mL) was added in the mixture. The target compound was purified by prep-TLC, dried to give Target 1d (10.0 mg, 11.9 % yield) as a white solid which was confirmed by 1H NMR (see Supplementary Notes: Spectroscopic Data for Generation of Analog 1d).

**Hydrogen–deuterium exchange mass spectrometry (HDX-MS) analysis.**
Hydrogen–deuterium exchange in PPM1D proteins (PPM1D$_{1-420}$, PPM1D$_{\Delta Flap}$, and PPM1D$_{\Delta Hinge}$) was monitored by incubating 27 pmoles of protein in storage buffer (50 mM Tris HCl pH 7.4, 500 mM NaCl, 10% glycerol, 10 mM MgCl2, 1 mM TCEP, H2O) with a 10-fold molar excess of GSK2830371 (270 pmoles) in DMSO, or 3.85% DMSO as a control. Due to the low solubility of the compound, the exact molar excess was unknown. However, the peptide spectra in the presence of the compound presented as single isotopic distributions (except where EX1 is indicated within the text), which suggests a large proportion of the protein was bound to the compound. The slow off-rate of the compound likely plays a role in this. The PPM1D:GSK/DMSO mix equilibrated for 15 min at room temperature, followed by the addition of a 15-fold excess of labeling buffer (20 mM HEPES, pD 7.5, 25 mM NaCl, 5 mM MgCl2, 0.1 mM TCEP, D2O) at 25 °C. Analysis of C-terminal truncations of PPM1D were performed without the DMSO and preincubation step. Labeling of PPM1D$_{\Delta loop}$, was performed with buffer composed of 20 mM HEPES, pD 7.5, 150 mM NaCl, 5 mM MgCl2, 0.1 mM TCEP in D2O due to solubility issues at low ionic strength (<150 mM NaCl). No difference in deuterium uptake was observed in PPM1D$_{1-420}$ in the two different salt concentrations. Labeling reactions were quenched with an equal volume of ice-cold quench buffer (0.8 M guanidine hydrochloride, 0.8% formic acid, H2O) at time points ranging from 10 s to 4 h. Undeuterated control samples were prepared for each of the proteins alone and in complex with GSK, where applicable, using the same procedures as outlined above and using labeling buffer made with H2O instead of D2O.

All samples were digested immediately after quench by injecting directly into an M-class nanoAcquity UPLC with HDX technology (Waters) for online digestion with an Enzymate BEH pepsin column [2.1 × 30 mm, 5 µm (Waters)] and UPLC peptide separation. The digestion chamber of the UPLC system was held at 15.0 ± 0.1 °C during analysis and the flow rate over the digestion column was 100 µL/min. The cooling chamber, which contained the chromatography separation, was held at 0.0 ± 0.1 °C for the entire analysis. Peptides were trapped and desalted on a VanGuard Pre-Column trap [2.1 mm × 5 mm, ACQUITY UPLC BEH C18, 1.7 µm (Waters)] for 3 min at 40 µL/min, eluted from the trap using a 5–35% gradient of acetonitrile over 6 min at a flow rate of 40 µL/min, and separated on an ACQUITY UPLC BEH C18, 1.8 µm, 1.0 mm × 100 mm column (Waters). The back pressure averaged 8500 psi at 0 °C and 5% acetonitrile:95% water. The error of determining the deuterium levels was ± 0.2 Da in this experimental setup based on. Therefore, differences ± 0.5 Da were considered meaningful. A wash cycle injection was performed between every two samples in which 100 µL wash solution (1.5 M GdnHcl, 4% acetonitrile, 0.8% formic acid) was injected over the digestion column and 5%-85% acetonitrile gradients repeated over the trap and analytical columns. Mass spectra were acquired using a Waters Xevo G2-XS QToF mass spectrometer equipped with a standard ESI source set to the following parameters: polarity ES+, capillary voltage, 3.0 kV, cone voltage, 50 V; collision energy, 4 V; cone gas flow, 90 L/h; source temperature, 80 °C; desolvation temperature, 175 °C; and desolvation gas flow, 600 L/h. StepWave parameters were adjusted as previously described[35]. The instrument scanned every 0.4 s from 100 to 2000 *m/z* in MS$^E$ mode, with collision energy ramp enabled from 20 V to 40 V. All comparison experiments

were done under identical experimental conditions such that deuterium levels are reported as relative. Back-exchange was calculated based on deuterium uptake observed in peptides within PPM1D that are solvent accessible, lack structure, and display these characteristics in their uptake profile (high deuterium incorporation that remains a flat line over 4 h time course). See uptake plots for peptides 45–71, 72–91 and 75–91 in Supplementary file "HDX Deuterium Uptake Plots".

Peptides were identified using PLGS 3.0.2 (Waters) using multiple replicates of undeuterated control samples. Raw MS data were imported into DynamX 3.0 (Waters). Peptides meeting the filtering criteria were further processed automatically by DynamX followed by manual inspection of all processing. Additional experimental details and processing parameters are outlined in Supplementary Table 4. The relative amount of deuterium in each peptide was determined by subtracting the centroid mass of the undeuterated form of each peptide from the deuterated form, at each time point, for each condition. Deuterium uptake values were used to calculate fractional percent uptake as follows:

$$\text{Fractional uptake \%} = \frac{\text{Da incorporated}}{\text{peptide length} - (\text{prolines}) - 1_{N-\text{termamideH}}}$$

Deuterium uptake values were also used to generate uptake difference maps, where differences are expressed in Daltons and differences greater than ±0.5 Da were considered significant based on the global $|\Delta HX|$ significance value of 0.405 Da, calculated as per Hageman and Weis[36]. Repeatability was determined as recommended by Masson et al., where standard deviation values from replicate experiments were calculated for each peptide at each time point[37]. The average of these SD values, σ, was 0.1 Da and repeatability was reported as 2σ or 0.2 Da. MS and analysis files are accessible through the UCSD MassIVE data repository and can be downloaded through the following link (ftp://massive.ucsd.edu/MSV000088711/). Supplementary Data 1 and 2 contain all of the data reported in this study.

**Analytical ultracentrifugation (AUC) analysis.** AUC experiments were done using an Optima ultracentrifuge (Beckman Instruments). Proteins were dialyzed at 4 ˚C prior to each experiment in 20 mM HEPES (pH 7.4), 100 mM NaCl, 5 mM MgCl2, 0.1 mM TCEP. Sedimentation velocity (SV-AUC) experiments were conducted using an An50 Ti rotor at 50,000 rpm and 20 °C following temperature equilibration for 1 h prior to the start of the experiment. Samples were prepared at 5 µM protein with either 2% DMSO alone or with 3x compound immediately before loading into dual-sector AUC cells. DMSO was necessary for compound solubility and added to all samples to ensure that all experiments were comparable; however, the inclusion of the organic solvent impeded our ability to establish an accurate molecular weight estimate and correction for Sw,20 values. We, therefore, present these SV-AUC data as relative sedimenting populations with their experimentally observed S values. Absorbance data were collected at 280 nm. SedFIT was used to model the sedimentation coefficient distributions [c(s) analysis] for each experiment[38,39]. Data were then refined using the c(s) with a conformational change model to better estimate the relative populations observed for PPM1D$_{1-420}$. SedNTerp was used to calculate protein parameters[40]. Images were generated using GUSSI[41].

**CD spectroscopy.** The protein was prepared for CD spectroscopy by dialyzing overnight into 10 mM phosphate, pH 7.4, 500 mM NaF. All data were collected on a Jasco J-815 CD spectrometer (Jasco, Inc.) equipped with a PTC-424S six-position Peltier-effect cell changer. The temperature was maintained at 25 °C for each experiment. Five scans were collected using a 1 mm path-length cell. Each spectrum was collected with a scanning speed of 2 nm/min, data interval time of 1 s, band width of 1 nm, and data pitch of 0.1 nm. For each spectrum collected, the contribution from the solvent was subtracted from the final spectrum.

**Reporting summary.** Further information on research design is available in the Nature Research Reporting Summary linked to this article.

## Data availability
The authors declare that all data supporting the findings of this study are available within the paper and its supplementary information files. MS and analysis files are accessible through the UCSD MassIVE data repository and can be downloaded through the following link [ftp://massive.ucsd.edu/MSV000088711/]. BLAST was used to obtain the structures of human PPM1A (3FXJ), human PPM1B (2P8E), and a PPM-type phosphatase (2I0O). Any additional information can be requested from the corresponding author. All data used in this study are available in the Source Data and Supplementary Data 1 and 2.

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

## Acknowledgements

Funding of this project was provided by U.S. National Institutes of Health (NIH) grants K12 CA087723 (to ASS); R01HL082945 and P01CA066996 (to B.L.E.). Further support was provided by the Evans Foundation (to B.L.E. and P.G.M.), the American Society of Hematology (to PGM), and the Howard Hughes Medical Institute (to B.L.E.). The authors would also like to express their thanks to Virendar Kaushik and other members of the CDoT leadership team at the Broad Institute for their support and guidance. The authors would also like to acknowledge Deerfield Ventures for their financial support of this work.

## Author contributions

P.G.M., M.S., J.A.M., W.J., C.G., and B.L.E. designed and conceived the study. P.G.M., M.S., J.A.M., C.G., Y.Q., S.I., Q.G., S.S., C.V., B.B., C.R., M.R., C.J., E.F., A.C., M.B., A.S.S., C.J.G., J.K., and A.J.C. performed the experiments and analyzed and interpreted the data. P.G.M., A.O.G., X.Y., F.P., and D.E.R. designed and generated the mutagenesis library, and analyzed the data from the mutagenesis screen. P.G.M., M.S., J.A.M., W.J., Y.Q., S.I., C.W.G., and B.L.E. drafted the manuscript.

## Competing interests

In the past 3 years, Dr. Miller has received consulting fees from Foundation Medicine. B.L.E. has received research funding from Celgene, Deerfield, Novartis, and Calico and consulting fees from GRAIL. He is a member of the scientific advisory board and shareholder for Neomorph Therapeutics, TenSixteen Bio, Skyhawk Therapeutics, and Exo Therapeutics. E.S.F. is a founder, science advisory board (SAB) member, and equity holder in Civetta Therapeutics, Jengu Therapeutics (board member), Neomorph Inc and an equity holder in C4 Therapeutics. E.S.F. is a consultant to Novartis, Sanofi, EcoR1 capital, Avilar and Deerfield. The Fischer lab receives or has received research funding from Astellas, Novartis, Voronoi, Ajax, and Deerfield. This work was financially supported by Deerfield Ventures; no employees of Deerfield Ventures were involved with the experimental design, experimental execution, data interpretation, figure generation, or manuscript preparation. The remaining authors declare no competing interest.
