## [Peer Review File · Nature Communications]

REVIEWER COMMENTS

Reviewer #1 (Remarks to the Author):

Dr Ebert and colleagues demonstrated that GSK2830371 a potent and selective allosteric inhibitor of PPM1D binds an allosteric site on PPM1D with high affinity. The authors relied on many biophysical and biochemical means to show that PPM1D exists in an equilibrium between two conformations that are defined by the movement of the flap domain, which is required for substrate recognition. They also identified that the hinge region was critical for switching between the two conformations and was directly implicated in the high-affinity binding of GSK2830371 to PPM1D.

The experiments are well planned and executed.

Although I do have few questions and suggestions to the authors

1. Was there any test of significance performed for the H/DX data for difference greater than 0.5 Da to be considered as a significant change?
2. I request the authors to provide the exchange pattern for each identified peptide and include error bars to the exchange points in an additional supplementary figure.
3. I would also suggest that if the error bars on the exchange time points are huge, to include additional 1 or 2 replicates of time points to get a more confident result.
4. What was the 100% exchange experiment that was used to calculate the back exchange rate?
5. The authors have mentioned about the use of Intact mass measurement and shown the peak area measurements for to detect dephosphorylation of T180 in full length p38 MAPK. Please provide the mass spectrum used for the measurement.

Reviewer #2 (Remarks to the Author):

The manuscript from Miller et al. describes the mechanism for interaction of the PPM1D phosphatase and its inhibitor GSK2830371, based on the modelled structure and saturation mutagenesis, and combination with HDX-MS, DSC, DSF, and ITC. As they described, they could not crystalize PPM1D protein despite of their effort. Therefore, they used homology modeling to obtain the PPM1D structure and compared the result with saturation mutagenesis analysis. They suggested that importance of the hinge region of PPM1D for the GSK2830371 inhibitory activity.

This work has some novelty in structural analysis. However, the findings are significant flaws in the lack of data and interpretations of results that preclude publication in Nature Communications.

Specific comments are indicated below:

Major points:

1. As shown in Supplemental Figure 4, the PPM1D Δ flap variant is inactive both for small molecule and protein substrates. In this case, it is unclear what information can be

obtained by comparing the differences in exposure of each region of the active phosphatase and the inactive variant. In addition, the results of the analysis for the interaction between the inactive variant and the inhibitor do not necessarily reflect the mechanism of inhibition of the active phosphatase.

2. The authors used GSK analogue 1d for their ITC experiments because the low solubility of GSK2830371 did not allow for the experiments.. However, there is no description of the synthesis of this compound and its PPM1D inhibitory activity. If the inhibitory activity of this analogue is significantly lower than that of GSK, the implications of the results of the ITC experiment become ambiguous. They should describe the synthesis of the compound and its PPM1D inhibitory activity.

3. In Supplement Figure 5D, the authors show the data of the ITC experiment using PPM1D variant and GSK2830371. This is very confusing. They should explain why the low solubility of GSK2830371 was not a problem in this experiment.

4. The authors emphasize the importance of the hinge region. However, this proposal is based on experiments using truncated variants substituted with a linker of short residues. Therefore, there is a lack of specific information of the important residue(s). Data should be provided to clarify which specific residue(s) in the hinge region are important.

5. The CD spectra of Figure 7C should be expressed as MRE (mean residue ellipticity) since the length of the PPM1D proteins is different. Otherwise, the authors cannot discuss the structural differences.

6. Line 214: The authors describe this sentence as "data not shown". This should not be allowed. They have to show the data for this.

Minor points

1. Raw data collected from DSC experiments in Figures 3C and 7C should be shown to clarify quality and accuracy.

2. Values of raw data for Figure 2C should be provided as an Excel file in Supplemental information.

3. Figure 1A: The region of Flap should be "219-295".

4. There are many typos in the METHODS section, including superscripts, subscripts, units, etc.

5. Line 453: Reference(s) for the PPM1D mutants of D314A, DGH(105-107) AAA, and MED(21-23)AAA should be listed.

6. Line 475: The values of SD is not shown in Figure 3C, D and F. Also, the actual numbers of experiments should be stated, not " $n \geq 3$ ".

Reviewer #3 (Remarks to the Author):

Thanks for the opportunity to review this interesting manuscript. In summary, the authors propose an evolutionarily informed model of PPM1D domain structure after being unable to crystalize the protein. They test this model using a saturation mutagenesis approach, which nicely reveals regions of the protein more and less sensitive to missense variation, as well as a position after which truncating mutations do not disrupt activity of the protein which corresponds nicely to where mutations cluster in cancer, implying loss of regulation. This is followed by biochemical characterisation and HDX-MS of the WT protein, unbound and bound to drug, as well as a few engineered constructs with deletions of key domains. These studies allow them to conclude with some confidence that the protein exists in two

structural states, which are distinguished by movement of a flap, and that a well-conserved, functionally essential hinge region is the site of GSK2830371 binding. They propose that GSK binding to this site “locks” the protein in an inactive conformation, thereby leading to highly specific inhibition of this particular phosphatase.

Overall, the paper’s conclusions seem reasonably sound and I’m inclined to recommend its publication after revisions, though I defer to those with expertise in HDX-MS to assess how those data contribute to the likelihood of the proposed model. Here, I focus specifically on the genetics and saturation mutagenesis approach. As implemented, the data seem fairly reliable for forming mechanistic predictions of protein function and overall the work represents a nice example of how a deep mutational scan can define functional elements of proteins. Although the finished product in Figure 2 looks well done, there are several areas in which more methodological detail and analytical consideration are needed before I can fully recommend this work for publication.

Therefore, please address the following points when resubmitting:

1. A minimal set of criteria is needed to thoroughly describe the implementation and performance of high-throughput saturation mutagenesis experiment to ensure reproducibility. Please update the methods / results to include all the following pieces:
 - a. Number of replicate library infections. If only 1, please justify this or do a second replicate.
 - b. Number of cells infected and sorted via FACS in each bin analyzed. Please also show the FACS data with gating strategy as a supplemental figure.
 - c. Complete PCR conditions used for targeted sequencing preparation.
 - d. Any custom code used for analysis must be included at time of publication, as well as a table of raw variant scores.
2. Taking a macro view of the data, the screen very clearly highlights regions intolerant to variation in PPM1D. However, it is unclear with what accuracy it predicts an individual variant’s functional effects. To get at this question, please include a plot comparing score distributions for all nonsense vs. synonymous variants (two distributions; can separate on before and after p.400 if easier to interpret that way). Adding all missense variants on top would help the reader understand the functional distribution present across the protein.
3. Was the screen ever performed in the presence of the inhibitor? This experiment may indicate resistance mutations that prevent drug binding at the hinge site if the proposed mechanism of action is correct, adding methodological novelty.
4. Some small points
 - a. Line 167 needs a reference, as it is unclear in reading it who has proposed this mechanism.
 - b. In methods, 2 g/ml of puromycin seems too high to be accurate.
 - c. I was unable to find references for the methods section. Please include.

Reviewer #4 (Remarks to the Author):

The manuscript entitled “Allosteric inhibition of PPM1D serine/threonine phosphatase via an altered conformational state” presents a lot of work to characterize the structure,

function, and inhibition of PPM1D, a serine/threonine phosphatase that regulates the DNA damage response and p53. Previous work had suggested that the allosteric inhibitor, GSK2830371 bound to the "flap region". In this work, the authors identify a second unique region, a "hinge" encompassing residues 155-166, also on the surface and near the "flap region" (residues 219-295) as the GSK2830371 binding site. Because the authors could not crystallize the protein, they build a homology model and validate it with saturation mutagenesis from which they learn a lot about important regions of the protein. The work in this paper is comprehensive, lots of biophysical measurements including melting temperature of the inhibitor-bound and free and many variants including flap deletion, hinge deletion, and C-terminal truncation. There are several issues that need to be addressed before this paper will be ready for publication.

1) The authors use HDX-MS to characterize the deuterium uptake of the various constructs and when inhibitor is bound. Although many HDX-MS practitioners do the same thing, it is really not OK to plot the peptides on the x-axis (cf Fig 4A, D, E, Fig 5A, Fig 6A, Fig 7D). This is because some regions have lots of overlapping peptides while others are not covered at all but the plots make it look like all regions are represented equally. It is better to use any one of a number of methods to determine the per-residue uptake and to plot these graphs with residue number on the x axis.

2) Line 219 the authors define the flap as 263-298 but the peptide they show is bimodal is 270-278. Do they have other bimodal peptides that cover the entire flap or is this a special feature of just these residues? In general, I found it difficult to relate their description of the results to the actual HDX-MS data in the figures because of a lack of specifics such as this one.

3) The Methods section was hastily prepared with several references missing and partial sentences, etc. The authors need to do a much better job of this in the final version.

4) I am not sure I agree with the authors interpretation (line 353) "Our HDX-MS data revealed EX1 kinetics in the flap region, suggesting that movement of the flap region mediates the equilibrium between these conformational states." It is equally likely that the flap region is responding to the hinge rather than the other way around. In fact, their data suggest this because deletion of the hinge causes the flap to move to the higher exchanging state much less (Fig. 4C). Binding of the inhibitor does the opposite, it shifts the flap to the higher exchanging state.

5) The Analytical Ultracentrifugation study is incomplete based on the possibilities outlined in 4). They are able to shift the equilibrium in both directions by deletion yet they don't analyze the hinge deletion (or at least they don't present the data). This important control should allow them to distinguish between whether the hinge deletion shifts the equilibrium between the two states.

6) The authors may consider merging the results and discussion section. It seemed they were looking for additional things to say in the discussion when most of the salient points were made in the results.

**RESPONSE TO REVIEWERS
NATURE COMMUNICATIONS**

**Allosteric inhibition of PPM1D serine/threonine phosphatase via an altered conformational state
(NCOMMS-21-39574)**

Reviewer #1 (Remarks to the Author):

Dr Ebert and colleagues demonstrated that GSK2830371 a potent and selective allosteric inhibitor of PPM1D binds an allosteric site on PPM1D with high affinity. The authors relied on many biophysical and biochemical means to show that PPM1D exists in an equilibrium between two conformations that are defined by the movement of the flap domain, which is required for substrate recognition. They also identified that the hinge region was critical for switching between the two conformations and was directly implicated in the high-affinity binding of GSK2830371 to PPM1D.

The experiments are well planned and executed.

We thank the reviewer for their positive comments and for highlighting the strengths of the paper.

Although I do have few questions and suggestions to the authors

1. Was there any test of significance performed for the H/DX data for difference greater than 0.5 Da to be considered as a significant change?

This is an excellent point raised by the reviewer. We apologize for not stating this clearly. In the original submission, we used the standard 2σ , or twice the standard deviation, of the average standard deviation of deuterium incorporation for all peptides to assess reproducibility of the measurements ($\sigma = 0.1$ Da; $2\sigma = 0.2$ Da). The significance level of 0.5 Da was originally chosen as a value in excess of this error, but we have since performed the global significance calculation as described by Hageman and Weis (Anal Chem 91, 8008-8016, 2019) showing the significance level to be 0.405 Da. These values are now presented in the Supplementary Excel file “HDXMS Deuterium Uptake Data” and summarized in Supplementary Table 5. We have also added the following to the Methods section:

Deuterium uptake values were also used to generate uptake difference maps, where differences are expressed in Daltons and differences greater than +/- 0.5 Da were considered significant based on the global $|\Delta\text{HX}|$ significance value of 0.405 Da, calculated as per Hageman and Weis. Repeatability was determined as recommended by Masson et al, where standard deviation values from replicate experiments were calculated for each peptide at each timepoint. The average of these SD values, σ , was 0.1 Da and repeatability was reported as 2σ or 0.2 Da. MS and analysis files are accessible through the UCSD Massive data repository and can be downloaded through the following link (<ftp://massive.ucsd.edu/MSV000088711/>).

2. I request the authors to provide the exchange pattern for each identified peptide and include error bars to the exchange points in an additional supplementary figure.

Per the reviewer’s request, uptake plots for all peptides are now included in the Supplementary file “HDX Deuterium Uptake Plots.”

3. I would also suggest that if the error bars on the exchange time points are huge, to include additional 1 or 2 replicates of time points to get a more confident result.

Thank you for this suggestion. As now shown in the updated figures and the standard deviation values in Supplementary Table 5 and the HDXMS Deuterium Uptake Data supplementary file, the error is small and therefore we do not believe requires additional replicates to provide more confidence in the results.

4. What was the 100% exchange experiment that was used to calculate the back exchange rate?

Thank you for this clarifying question. Back-exchange on the LC/MS system was estimated using peptides from the loop region of PPM1D (residues 39-95), that were predicted to be unstructured (Fig 1A) and showed high relative deuterium uptake at early timepoints with no increase in uptake over the course of the entire experiment (see peptide plots covering residues 45-91 on page 1 of the HDX Deuterium Uptake Plots file). These data support that the loop was completely exposed to deuterium in the exchange experiments. The retention of deuterium in peptides spanning this region ranged from 60-70%, so we considered the back exchange of our system to be ~35%. Furthermore, as these experiments are comparison studies, deuterium uptake is reported as relative and absolute deuterium content is not necessary. This has been added to the methods section.

Back-exchange was calculated based on deuterium uptake observed in peptides within PPM1D that are solvent accessible, lack structure, and display these characteristics in their uptake profile (high deuterium incorporation that remains a flat line over the four-hour timecourse). See uptake plots for peptides 45-71, 72-91 and 75-91 in Supplementary file “HDX Deuterium Uptake Plots”.

5. The authors have mentioned about the use of Intact mass measurement and shown the peak area measurements for to detect dephosphorylation of T180 in full length p38 MAPK. Please provide the mass spectrum used for the measurement.

We have added the mass spectra for substrate (p38-pT180) and product (p38-T180) proteins as requested by the reviewer to Supplementary Figure 2C.

Reviewer #2 (Remarks to the Author):

The manuscript from Miller et al. describes the mechanism for interaction of the PPM1D phosphatase and its inhibitor GSK2830371, based on the modelled structure and saturation mutagenesis, and combination with HDX-MS, DSC, DSF, and ITC. As they described, they could not crystalize PPM1D protein despite of their effort. Therefore, they used homology modeling to obtain the PPM1D structure and compared the result with saturation mutagenesis analysis. They suggested that importance of the hinge region of PPM1D for the GSK2830371 inhibitory activity.

This work has some novelty in structural analysis. However, the findings are significant flaws in the lack of data and interpretations of results that preclude publication in Nature Communications.

Thank you for highlighting the main findings of our work. We believe we have addressed the issues you outlined below.

Specific comments are indicated below:

Major points:

1. As shown in Supplemental Figure 4, the PPM1D_{Δflap} variant is inactive both for small molecule and protein substrates. In this case, it is unclear what information can be obtained by comparing the differences in exposure of each region of the active phosphatase and the inactive variant. In addition, the results of the

analysis for the interaction between the inactive variant and the inhibitor do not necessarily reflect the mechanism of inhibition of the active phosphatase.

We apologize for any confusion caused here. The key point we are trying to convey here relates to the interaction of GSK2830371 with PPM1D_{Δflap}. The fact that it is catalytically inactive did not preclude it from binding to GSK2830371 (as shown in multiple biophysical assays) and revealing changes in the HDX-MS deuterium uptake profile. Therefore, the key conclusion is that the flap region of PPM1D is not required for high-affinity binding of GSK2830371 or to induce dynamic changes of the protein upon compound binding. We believe that these data inform our overall working model of inhibition regarding the role of the hinge region in modulating movement of the flap domain with GSK2830371 binding and inhibiting catalytic function.

2. The authors used GSK analogue 1d for their ITC experiments because the low solubility of GSK2830371 did not allow for the experiments.. However, there is no description of the synthesis of this compound and its PPM1D inhibitory activity. If the inhibitory activity of this analogue is significantly lower than that of GSK, the implications of the results of the ITC experiment become ambiguous. They should describe the synthesis of the compound and its PPM1D inhibitory activity.

Thank you for pointing out this oversight. Analogue 1d had a K_d in SPR of 2.7nM and an IC₅₀ in the p38 assay of 22.4 nM. These data are now included in the text and in Supplementary Figure 3B-C. Additionally we have included the complete synthesis of analog 1d in the methods section and Supplementary Figure 3D. We have also added the following to the results section:

As the relatively low solubility of GSK2830371 precluded isothermal titration calorimetry (ITC), we generated a structural analog of GSK2830371 (analog 1d) with comparable potency and binding affinity (K_d = 2.7nM in SPR and IC₅₀=22.4 nM in p38 MAPK assay) to GSK2830371 but significantly higher solubility (>100 μM) (Figure 3E and Supplementary Figures 3A-D).

'3. In Supplement Figure 5D, the authors show the data of the ITC experiment using PPM1D variant and GSK2830371. This is very confusing. They should explain why the low solubility of GSK2830371 was not a problem in this experiment.

We apologize for the confusion. The ITC experiments were performed with analogue 1d. We have updated the Figure 3F legend:

(F) The thermodynamic profile of the interaction of the GSK2830371 analogue (1d) with PPM1D₁₋₄₂₀ as measured by isothermal calorimetry.

4. The authors emphasize the importance of the hinge region. However, this proposal is based on experiments using truncated variants substituted with a linker of short residues. Therefore, there is a lack of specific information of the important residue(s). Data should be provided to clarify which specific residue(s) in the hinge region are important.

The reviewer correctly points out that while we were able to identify the hinge region of the protein as a mediator of high affinity binding of GSK2830371, our HDX-MS data does not have the resolution to define a specific residue. This is an important question that will be the focus of ongoing work and we believe that it is more likely than not that multiple residues in this region are required for drug binding.

5. The CD spectra of Figure 7C should be expressed as MRE (mean residue ellipticity) since the length of the PPM1D proteins is different. Otherwise, the authors cannot discuss the structural differences.

Thank you for pointing this out. We have updated the figure.

6. Line 214: The authors describe this sentence as “data not shown”. This should not be allowed. They have to show the data for this.

“This included the N-terminal peptides spanning residues 5-12, supporting the homology model and experimental observations that N-terminal deletions of PPM1D lead to insoluble protein (data not shown).”

Thank you for highlighting this in the text. The reason we put “data not shown” was because the insolubility of the protein precluded our ability to perform further protein characterization. As a result, there are no figures such as gels that would be helpful to the reader as they would all be blank. We have updated the text to be more specific:

Furthermore, loss of the N-terminal residues results in insoluble, aggregated recombinant protein which precluded further characterization of the protein lacking the N-terminus

Minor points

1. Raw data collected from DSC experiments in Figures 3C and 7C should be shown to clarify quality and accuracy.

Please see the raw data now presented in the Supplementary Figure 2D and Supplementary Figure 6B.

2. Values of raw data for Figure 2C should be provided as an Excel file in Supplemental information.

Thank you for highlighting this missing data. We have now generated Supplemental Table 4 that includes the raw data from the saturation mutagenesis experiment shown in figure 2C.

3. Figure 1A: The region of Flap should be “219-295”.

Thank you for pointing this out. We have fixed the figure as requested.

4. There are many typos in the METHODS section, including superscripts, subscripts, units, etc.

Our apologies for these. We have reviewed the methods section and fixed the typos that the reviewer is referring to.

5. Line 453: Reference(s) for the PPM1D mutants of D314A, DGH(105-107) AAA, and MED(21-23)AAA should be listed.

Thank you for highlighting this oversight. We have added these references to the manuscript.

6. Line 475: The values of SD is not shown in Figure 3C, D and F. Also, the actual numbers of experiments should be stated, not “ $n \geq 3$ ”.

Sorry for not listing this out. Please see the new figures 3C, D and F with the respective SD and also the exact number of experiments performed for each figure in the figure legend.

Reviewer #3 (Remarks to the Author):

Thanks for the opportunity to review this interesting manuscript. In summary, the authors propose an evolutionarily informed model of PPM1D domain structure after being unable to crystalize the protein. They test this model using a saturation mutagenesis approach, which nicely reveals regions of the protein more and less sensitive to missense variation, as well as a position after which truncating mutations do not disrupt activity of the protein which corresponds nicely to where mutations cluster in cancer, implying loss of regulation. This is followed by biochemical characterization and HDX-MS of the WT protein, unbound and bound to drug, as well as a few engineered constructs with deletions of key domains. These studies allow them to conclude with some confidence that the protein exists in two structural states, which are distinguished by movement of a flap, and that a well-conserved, functionally essential hinge region is the site of GSK2830371 binding. They propose that GSK binding to this site “locks” the protein in an inactive conformation, thereby leading to highly specific inhibition of this particular phosphatase.

Overall, the paper’s conclusions seem reasonably sound and I’m inclined to recommend its publication after revisions, though I defer to those with expertise in HDX-MS to assess how those data contribute to the likelihood of the proposed model. Here, I focus specifically on the genetics and saturation mutagenesis approach. As implemented, the data seem fairly reliable for forming mechanistic predictions of protein function and overall the work represents a nice example of how a deep mutational scan can define functional elements of proteins. Although the finished product in Figure 2 looks well done, there are several areas in which more methodological detail and analytical consideration are needed before I can fully recommend this work for publication.

We thank the reviewer for their thoughtful summary of the main conclusions and strengths of our work.

Therefore, please address the following points when resubmitting:

1. A minimal set of criteria is needed to thoroughly describe the implementation and performance of high-throughput saturation mutagenesis experiment to ensure reproducibility. Please update the methods / results to include all the following pieces:

We apologize for this oversight and have included all of the below in the text in addition to the in-line responses below.

a. Number of replicate library infections. If only 1, please justify this or do a second replicate.

Two replicates were performed which included independent infection, treatment, sorting, and sequencing. The methods section now includes:

The entire screen was performed in duplicate. After two independent rounds of infection of the K562-TP53wild-type -CDKN1A-GFP cells at a low multiplicity of infection, cells were selected with 2 ug/ml puromycin (ThermoFisher, #A1113802) for two days.

b. Number of cells infected and sorted via FACS in each bin analyzed. Please also show the FACS data with gating strategy as a supplemental figure.

One hundred million cells were infected in each replicate with the goal of achieving approximately 1000-1500 cells per variant cDNA after puromycin selection given ~25% infection efficiency ((100,000,000*0.25)/16,377 ~1,500 cells/cDNA). The FACS sorting strategy is now included in Supplemental Figure 1B in which approximately the top and bottom 15-20% of cells were sorted.

c. Complete PCR conditions used for targeted sequencing preparation.

The PCR primers used were Forward 5' ATTCTCCTTGGAAATTTGCCCTT 3' and Reverse 5' CATAGCGTAAAAGGAGCAACA 3'. The PCR cycling program was as follows: 98C 30 seconds; 35 cycles of [98C 10 seconds; 64C 30 seconds; 72C 2 minutes]; 72C 2 minutes; 4C hold. This has been added to the methods section.

d. Any custom code used for analysis must be included at time of publication, as well as a table of raw variant scores.

We have now included a table of raw variant scores (Supplementary Table 4) that were used to generate the heatmap in Figure 2C. The methodology for generating the data, including the code used, has been previously described and we have added additional references for clarity.

2. Taking a macro view of the data, the screen very clearly highlights regions intolerant to variation in PPM1D. However, it is unclear with what accuracy it predicts an individual variant's functional effects. To get at this question, please include a plot comparing score distributions for all nonsense vs. synonymous variants (two distributions; can separate on before and after p.400 if easier to interpret that way). Adding all missense variants on top would help the reader understand the functional distribution present across the protein.

Thank you for this suggestion. As the reviewer suggested, it can be challenging to make calls of specific residues with the same degree of certainty that can be applied to a stretch of amino acids. However, as requested we have added Supplemental Figure 1C which plots the individual raw data for silent, missense (averaged over all missense), and nonsense mutations in addition to an overlay of the smooth-fit curves for all three. The silent mutations, as expected, fall largely around 0-0.2 with few outliers, whereas there is more variability (particularly regional variability) in the missense mutations. In contrast, the nonsense mutations are largely of negative values, an expected result as truncations prior to amino acid 400 are expected to impair protein function. As the reviewer points out, this data depiction does nicely show that truncating mutations toward the end of the protein after amino acid 400 actually increase protein activity, consistent with the mutational spectrum observed in human disease and the presence of a c-terminal degradation domain in this region.

3. Was the screen ever performed in the presence of the inhibitor? This experiment may indicate resistance mutations that prevent drug binding at the hinge site if the proposed mechanism of action is correct, adding methodological novelty.

This is an excellent question and a major area of effort moving forward. We have not performed this screen but are hoping to do so in the near future with the goal of identifying specific residues of the protein, particularly in the hinge region that may be important for drug inhibition and binding.

4. Some small points

a. Line 167 needs a reference, as it is unclear in reading it who has proposed this mechanism.

Thank you for pointing this out. We have added the appropriate reference (Gilmartin et al Nat Chem Bio 2014) to the text.

b. In methods, 2 g/ml of puromycin seems too high to be accurate.

Thank you for seeing this! We adjusted the methods text to 2ug/ml.

c. I was unable to find references for the methods section. Please include.

Thank you for highlighting this and we apologize for the confusion. The references for the methods section are linked to the main methods.

Reviewer #4 (Remarks to the Author):

The manuscript entitled “Allosteric inhibition of PPM1D serine/threonine phosphatase via an altered conformational state” presents a lot of work to characterize the structure, function, and inhibition of PPM1D, a serine/threonine phosphatase that regulates the DNA damage response and p53. Previous work had suggested that the allosteric inhibitor, GSK2830371 bound to the “flap region”. In this work, the authors identify a second unique region, a “hinge” encompassing residues 155-166, also on the surface and near the “flap region” (residues 219-295) as the GSK2830371 binding site. Because the authors could not crystallize the protein, they build a homology model and validate it with saturation mutagenesis from which they learn a lot about important regions of the protein. The work in this paper is comprehensive, lots of biophysical measurements including melting temperature of the inhibitor-bound and free and many variants including flap deletion, hinge deletion, and C-terminal truncation. There are several issues that need to be addressed before this paper will be ready for publication.

1) The authors use HDX-MS to characterize the deuterium uptake of the various constructs and when inhibitor is bound. Although many HDX-MS practitioners do the same thing, it is really not OK to plot the peptides on the x-axis (cf Fig 4A, D, E, Fig 5A, Fig 6A, Fig 7D). This is because some regions have lots of overlapping peptides while others are not covered at all but the plots make it look like all regions are represented equally. It is better to use any one of a number of methods to determine the per-residue uptake and to plot these graphs with residue number on the x axis.

We agree with the reviewer’s concern that plotting the data in this format can be misleading. However, many methods calculating the per-residue uptake require a significant number of overlapping peptides to determine a result with high confidence, which we did not observe in many cases. To address this issue, we have included coverage maps for each experiment in the Supplementary Excel file, HDXMS Deuterium Uptake Data. Further, we have added vertical tick marks to the x-axes to indicate regions missing coverage. We have updated the figure legends describing the x-axes for figures 4 – 7 and Supp. Figs 5 and 6 to include:

Vertical tick marks between peptides are shown to indicate regions of missing coverage.

2) Line 219 the authors define the flap as 263-298 but the peptide they show is bimodal is 270-278. Do they have other bimodal peptides that cover the entire flap or is this a special feature of just these residues? In general, I found it difficult to relate their description of the results to the actual HDX-MS data in the figures because of a lack of specifics such as this one.

We apologize for the lack of clarity in our explanation of the HDX-MS data. We have clarified line 219 to explicitly indicate which residues comprise the entire flap region, versus those that exhibit bimodal distributions. 270-278 is a peptide that was consistently followed in every experiment and was chosen for clarity. We have included a column in the Supplementary Excel file, HDXMS Deuterium Uptake Data, to indicate which peptides displayed bimodal distributions in this region.

3) The Methods section was hastily prepared with several references missing and partial sentences, etc. The authors need to do a much better job of this in the final version.

We are very sorry for the errors that were present in the methods section. We have updated it to address the oversights pointed out by the reviewer.

4) I am not sure I agree with the authors interpretation (line 353) “Our HDX-MS data revealed EX1 kinetics in the flap region, suggesting that movement of the flap region mediates the equilibrium between these conformational states.” It is equally likely that the flap region is responding to the hinge rather than the other way around. In fact, their data suggest this because deletion of the hinge causes the flap to move to the higher exchanging state much less (Fig. 4C). Binding of the inhibitor does the opposite, it shifts the flap to the higher exchanging state.

Thank you for this comment and pointing out that our interpretation was misleading. We agree with the reviewer’s interpretation that the hinge is mediating the movement of the flap. We have updated the text to reflect that the flap movement is a response to the position and flexibility of the hinge.

At baseline, PPM1D exists in two conformations. Our HDX-MS data revealed bimodal distributions in the flap region, suggesting that movement of the flap region dictates the equilibrium between these conformational states and that this flap movement is a response to the position and flexibility of the hinge.

5) The Analytical Ultracentrifugation study is incomplete based on the possibilities outlined in 4). They are able to shift the equilibrium in both directions by deletion yet they don’t analyze the hinge deletion (or at least they don’t present the data). This important control should allow them to distinguish between whether the hinge deletion shifts the equilibrium between the two states.

As suggested by the reviewer, we performed this experiment but the hinge deletion was not tolerated in the buffer conditions used (100mM NaCl and precipitated out) so we were unable to perform AUC experiment.

6) The authors may consider merging the results and discussion section. It seemed they were looking for additional things to say in the discussion when most of the salient points were made in the results.

We appreciate the reviewer’s suggestion but believe that separation of the discussion from the results as currently structured allows us to more clearly separate the data from our interpretations and efforts to contextualize the results.

REVIEWERS' COMMENTS

Reviewer #1 (Remarks to the Author):

The authors have answered my queries satisfactorily. It is exciting work and planned and executed meticulously. Therefore, I recommend the paper be published.

Reviewer #2 (Remarks to the Author):

The authors added new data including deuterium exchange. They corrected and revised the manuscript in response to criticisms from the referees. The revision has satisfactorily addressed referees' comments.
Now I think that the revised version is acceptable for publication in Nature Communications.

Reviewer #4 (Remarks to the Author):

The authors have addressed all of my concerns and the manuscript is now acceptable.

RESPONSE TO REVIEWERS

NATURE COMMUNICATIONS

Allosteric inhibition of PPM1D serine/threonine phosphatase via an altered conformational state (NCOMMS-21-39574)

Reviewer #1 (Remarks to the Author):

The authors have answered my queries satisfactorily. It is exciting work and planned and executed meticulously. Therefore, I recommend the paper be published.

Thank you for these positive comments and your excitement for our work.

Reviewer #2 (Remarks to the Author):

The authors added new data including deuterium exchange. They corrected and revised the manuscript in response to criticisms from the referees. The revision has satisfactorily addressed referees' comments. Now I think that the revised version is acceptable for publication in Nature Communications.

Thank you.

Reviewer #4 (Remarks to the Author):

The authors have addressed all of my concerns and the manuscript is now acceptable.

Thank you.